∂ | **Open Peer Review** | Ecology | Research Article

# Unveiling the co-phylogeny signal between plunderfish *Harpagifer* spp. and their gut microbiomes across the Southern Ocean

Guillaume Schwob,[1,2,3] Léa Cabrol,[1,3,4] Thomas Saucède,[5] Karin Gérard,[1,6,7] Elie Poulin,[1,2,3] Julieta Orlando[1,2]

**ABSTRACT** Understanding the factors that sculpt fish gut microbiome is challenging, especially in natural populations characterized by high environmental and host genomic complexity. However, closely related hosts are valuable models for deciphering the contribution of host evolutionary history to microbiome assembly, through the underscoring of phylosymbiosis and co-phylogeny patterns. Here, we propose that the recent diversification of several *Harpagifer* species across the Southern Ocean would allow the detection of robust phylogenetic congruence between the host and its microbiome. We characterized the gut mucosa microbiome of 77 individuals from four field-collected species of the plunderfish *Harpagifer* (Teleostei, Notothenioidei), distributed across three biogeographic regions of the Southern Ocean. We found that seawater physicochemical properties, host phylogeny, and geography collectively explained 35% of the variation in bacterial community composition in *Harpagifer* gut mucosa. The core microbiome of *Harpagifer* spp. gut mucosa was characterized by a low diversity, mostly driven by selective processes, and dominated by a single *Aliivibrio* Operational Taxonomic Unit (OTU) detected in more than 80% of the individuals. Nearly half of the core microbiome taxa, including *Aliivibrio*, harbored co-phylogeny signal at microdiversity resolution with host phylogeny, indicating an intimate symbiotic relationship and a shared evolutionary history with *Harpagifer*. The clear phylosymbiosis and co-phylogeny signals underscore the relevance of the *Harpagifer* model in understanding the role of fish evolutionary history in shaping the gut microbiome assembly. We propose that the recent diversification of *Harpagifer* may have led to the diversification of *Aliivibrio*, exhibiting patterns that mirror the host phylogeny.

**IMPORTANCE** Although challenging to detect in wild populations, phylogenetic congruence between marine fish and its microbiome is critical, as it highlights intimate associations between hosts and ecologically relevant microbial symbionts. Our study leverages a natural system of closely related fish species in the Southern Ocean to unveil new insights into the contribution of host evolutionary trajectory on gut microbiome assembly, an underappreciated driver of the global marine fish holobiont. Notably, we unveiled striking evidence of co-diversification between *Harpagifer* and its microbiome, demonstrating both phylosymbiosis of gut bacterial communities and co-phylogeny of some specific bacterial symbionts, mirroring the host diversification patterns. Given *Harpagifer*'s significance as a trophic resource in coastal areas and its vulnerability to climatic and anthropic pressures, understanding the potential evolutionary interdependence between the hosts and its microbiome provides valuable microbial candidates for future monitoring, as they may play a pivotal role in host species acclimatization to a rapidly changing environment.

Address correspondence to Guillaume Schwob, gschwob@institutobase.cl.

The authors declare no conflict of interest.

See the funding table on p. 16.

**KEYWORDS**  microbiome, phylosymbiosis, co-phylogeny, co-diversification, *Harpagifer*, *Aliivibrio*, teleost, Southern Ocean

The implication of the microbiome in facilitating or responding to host evolutionary processes is one of the burning points of holobiont studies (1–3). Under the holobiont concept, the reciprocal evolution of host and microbiome genomes, namely, co-evolution, is associated with several key life history traits such as obligate symbiosis, vertical inheritance, metabolic cooperation, reproduction control, and co-diversification (4–6). Co-diversification, defined as the parallel and synchronized diversification of the host and symbiont lineages through a history of constant association, constitutes the most investigated trait to get evidence of potential co-evolution in natural macroorganisms' populations (5, 7–9). However, demonstrating co-diversification between hosts and complex microbiomes in natural populations remains methodologically challenging because signal might be weak and/or overwhelmed by environmental factors, and insufficient genomic data hinder the determination of divergence timing and the acquisition of well-resolved phylogenies (8, 9). Two empirical patterns are classically recognized, known as phylosymbiosis and co-phylogeny, to evidence the impact of the evolutionary interactions on holobiont assemblage (9–11). Phylosymbiosis refers to a congruence pattern between the phylogeny of host species and the clustering of microbial community structure, observed at one moment in time and space (10). However, this pattern does not imply any stable evolutionary association between a host and its microbiome along time (12). Practically, phylosymbiosis is reflected by higher similarity of microbial communities within the same host species than between different host species and by genetic differences among hosts that are consistent with the compositional differences in their microbiomes (13). By contrast, co-phylogeny involves parallel evolutionary history of host species and specific microbial symbionts (5). A co-phylogenetic signal is elucidated by congruent topologies of host species and specific symbionts phylogenies, by which interacting partners shared similar positions in their respective trees (9). The screening for phylosymbiosis and co-phylogeny signals in complex and uncharacterized holobionts has led to the identification of specific microbes, with potentially highly relevant ecological role in the host (14–16).

Detecting robust phylosymbiosis and co-phylogeny signals in wild species populations is challenging due to the complexity of natural holobiont systems related to uncontrolled sources of microbial variability (10, 17). To tackle this issue, several approaches have been proposed, such as focusing on the core microbiome (i.e., common microbial taxa across diverse environments) (17, 18) and on the mucosa resident microbiome (i.e., autochthonous and supposedly temporally stable in the intestinal mucus) (3, 13, 19). These host-specific sub-communities, less impacted by external environment and diet (19, 20), likely fulfill critical functions for the host and contribute to its fitness and evolution (4, 13) and hence are more susceptible to present co-phylogeny patterns (12). Moreover, in anciently diverged hosts species with long-term evolution from the last common ancestor, the phylosymbiosis signal can be blurred by evolutionary history events (e.g., host-swap, symbiont extinction, or phylogenetically non-congruent symbiont speciation events) (10, 16). Several authors advocated working with recently diverged and thus genetically closely related species, characterized by stronger phylosymbiosis pattern, to better delineate the effects of evolutionary history and ecology of the host on microbiome assembly (17, 21, 22).

The fish fauna of the Southern Ocean (SO) is dominated by the perciform suborder Notothenioidei (teleost fishes), which constitute 90% of the fish biomass and up to 77% of the species diversity of the Antarctic continental shelf (23). To date, the microbial communities associated to the notothenioids have received very little attention and have been mainly characterized through cultural-dependent methods (24–26). The gut microbiome of only two species from West Antarctic Peninsula, *Notothenia coriiceps* and *Chaenocephalus aceratus*, has been analyzed through 16S rRNA gene-based clone libraries, revealing a low microbial richness and a strong dominance of the family

*Vibrionaceae* represented by *Vibrio* and *Photobacterium* genera (23). However, none of these studies addressed co-diversification within the notothenioid holobiont.

Within the Notothenioidei suborder, the monogeneric family of the *Harpagiferidae* comprises 12 nominal species, each one distributed in a specific region of the SO, such as *Harpagifer antarcticus* along West Antarctic Peninsula (27), *H. georgianus* in the South Georgia Islands (28), *H. bispinis* in Patagonia (29), and *H. kerguelensis* in Kerguelen Islands (30). These species are stenothermic, demonstrating an adaptation to cold waters, and have been identified as susceptible to the impacts of climate change (e.g., seawater temperature increase) and anthropogenic perturbation (i.e., microplastic contamination) in the Southern Ocean (31, 32). In this context, exploring potential evolutionary interdependence between *Harpagifer* and its specific symbionts could unveil microbial candidates for future monitoring. These symbionts may play a crucial role in either contributing to or limiting the acclimatization of host species to a rapidly changing environment (33). To date, only one published study delved into the evolutionary history of the genus *Harpagifer*, unveiling a recent differentiation between *H. antarcticus* and the South American species *H. bispinis*, estimated at 1.2–0.8 million years ago (34). This recent divergence, as observed for other species in the current study, makes *Harpagifer* an interesting model to explore co-diversification hypothesis. Despite relatively contrasting environmental conditions due to their distinct geographical distribution, these closely related species have the same trophic positioning and live in intertidal and shallow subtidal habitats (35, 36). Consequently, we anticipate the detection of shared microbial taxa, conforming the core of the gut mucosa microbiome ("GMM" hereafter) among *Harpagifer* species. The slow evolution of the 16S rRNA gene marker of bacteria, at 1%–2% per 50 Myr on average (37), precludes exploring pattern of co-phylogeny among symbionts and their host species that have diverged in shorter timescales when considering the classical 97% similarity bacterial Operational Taxonomic Units (OTUs) (38). Therefore, the co-phylogenetic signal of *Harpagifer* species and their microbiomes will be explored at a microdiversity level (39). By characterizing through 16S rRNA gene sequencing the GMM of wild-caught individuals from four species of *Harpagifer*, we aim (i) to evaluate the contribution of host identity and phylogeny on gut microbiome composition compared to the environment and the geography and (ii) to test the hypothesis of a co-phylogenetic signal between *Harpagifer* species and shared members of their gut microbiomes (i.e., core microbial taxa).

## MATERIALS AND METHODS

### *Harpagifer* spp. individuals sampling and dissection

Individuals of the fish species *H. bispinis*, *H. georgianus*, *H. kerguelensis*, and *H. antarcticus* were sampled between 2015 and 2021 from 12 localities of the SO, including two localities in the Chilean Patagonia (PAT1 and PAT2), five localities in South Georgia, one locality in the Kerguelen Islands (KER), and four localities in the West Antarctic Peninsula (WAP1–WAP4), respectively (Fig. 1; Table 1). Individuals were euthanized using buffered seawater containing >250 mg/L benzocaine (BZ-20, Veterquimica) and were then conserved in absolute ethanol at 4°C until dissection. Once at the laboratory, the *Harpagifer* individuals were aseptically dissected to remove the intestinal content, and the gut mucosa were gently rinsed with nuclease-free sterile water (Winkler) and stored at −20°C until DNA extraction.

### Genomic DNA extraction, PCR amplification, and amplicon sequencing analysis

DNA was extracted from gut mucosa samples using the DNeasy PowerSoil Pro Kit (Qiagen), with a preliminary incubation at 65°C for 10 min followed by a homogenization step using a FastPrep-24 bead beating grinder (MP Biomedicals). The V3–V4 region of the bacterial 16S rRNA gene was amplified by touchdown PCR using the modified

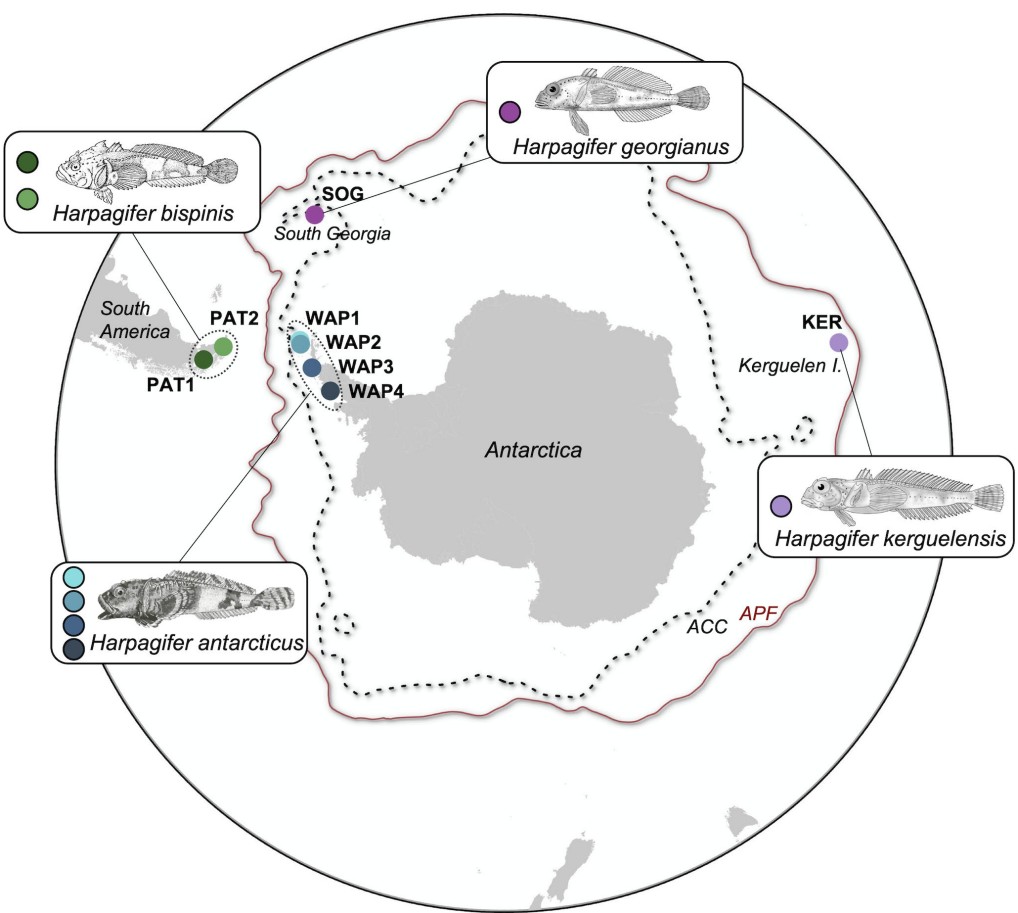

**FIG 1** Sampled *Harpagifer* sp. populations across the Southern Ocean. The designation of sampling sites is detailed in Table 1. Both Antarctic Circumpolar Current (ACC) and Antarctic Polar Front (APF) are represented on the map.

Bakt_341F/Bakt_805R primer pair (40). PCR products were purified and sequenced using the paired-end sequencing technology (2 × 300 bp) on the Illumina MiSeq Sequencer at the University of Wisconsin–Madison Biotechnology Center's DNA Sequencing Facility (USA). Reads of 16S rRNA gene were processed through MOTHUR (v1.48.0) (41), using the trimming criteria detailed in Schwob et al. (42). Processed sequences were clustered into OTUs at 97% similarity threshold similarity, discarding the OTUs conformed by a single sequence.

The host mitochondrial COI gene was amplified from the same DNA samples, using the FISH-F2/HCO2198-R primer pair (43, 44). Amplicons were purified and sequenced in both directions at Macrogen, Inc. (South Korea), using Sanger technology. Sequences of COI gene from the *Harpagifer* individuals were aligned, and polymorphic sites were visually checked in PROSEQ (45).

## Host genetic diversity, genetic distance, phylogeographic structure, and phylogenetic reconstruction

We estimated levels of polymorphism in *H. bispinis*, *H. georgianus*, *H. kerguelensis*, and *H. antarcticus* for the COI data sets in the ARLEQUIN software (v3.5.2) using standard diversity indices: haplotype number, number of polymorphic sites, haplotype diversity, average number of pairwise differences, and nucleotide diversity. Pairwise distances (p-distances) between species were calculated using Kimura-2-parameter. The statistical significance of genetic distances was assessed by conducting 10,000 permutations of individuals between the different species. The haplotype network was reconstructed

**TABLE 1** Sampling locations of *Harpagifer* species and overview of amplicon sequencing data[a,b]

| Species | Region | Locality | Date | GPS coordinates | N | N seq. (Relat. Abund.) |
|---|---|---|---|---|---|---|
| *Harpagifer antarcticus* | West Antarctic Peninsula (**WAP**) | Antarctic China Base "Great Wall," Fildes Peninsula, King George Island, South Shetland Islands (**WAP1**) | 01-2020 | 62°12′28.20″S 58°57′38.00″W | 10 | 318,876 (10.7) |
| | | Antarctic Chilean Base "Captain Arturo Prat," Iquique Cove, Greenwich Island, South Shetland Islands (**WAP2**) | 01-2018 | 62°28′43.84″S 59°40′14.90″W | 10 | 490,478 (16.4) |
| | | Antarctic Chilean Base "Yelcho," South Bay, Doumer Island, Antarctic Peninsula (**WAP3**) | 01-2018 | 64°52′33.13″S 63°35′00.96″W | 12 | 373,151 (12.5) |
| | | Horseshoe, Marguerite Bay, Antarctic Peninsula (**WAP4**) | 01-2022 | 67°53′32.82″S 67°24′17.34″W | 10 | 247,211 (8.3) |
| *Harpagifer kerguelensis* | Kerguelen Islands (**KER**) | Port-aux-Français, Gulf of Morbihan, French Southern and Antarctic Lands | 12-2015 | 49°21′13.32″S 70°13′56.76″E | 11 | 255,340 (8.5) |
| *Harpagifer georgianus* | South Georgia Island (**SOG**) | Esbensen Bay | 11-2021 | 54°52′0.96″S 35°57′46.98″W | 7 | 297,991 (10.0) |
| | | Gold Harbour | 11-2021 | 54°36′59.16″S 35°55′15.90″W | | |
| | | Albatross Island, Sunset Fjord | 11-2021 | 54°1′37.56″S 37°19′26.40″W | | |
| | | Sector Godthul | 11-2021 | 54°17′24.12″S 36°17′6.30″W | | |
| | | Luck Point, Bay of Isles | 11-2021 | 54°3′37.92″S 37°15′54.90″W | | |
| *Harpagifer bispinis* | Chilean Patagonia (**PAT**) | Sector "Aguas Frescas," Punta Arenas, Strait of Magellan, Magallanes and Chilean Antarctica Region (**PAT1**) | 02-2021 | 53°26′00.23″S 70°58′24.37″W | 11 | 558,754 (18.6) |
| | | Sector "Corrales Viejos," Puerto Williams, Navarino Island, Magallanes and Chilean Antarctica Region (**PAT2**) | 04-2021 | 54°55′56.89″S 67°28′20.89″W | 10 | 448,484 (15.0) |

[a]N, number of individuals per species; N seq., total number of sequences after bioinformatic processing; Relat. Abund., relative abundance in the whole data set.
[b]Site abbreviations are presented in boldface.
[c]Given the low number of individuals per locality within the South Georgia Island, the samples from the five localities were pooled.

using the median joining method using Populational Analysis with Reticulate Trees software (PopART, v1.7.0) (46).

For phylosymbiosis analysis, the phylogenetic tree of the 77 individual COI sequences was reconstructed using the PhyML algorithm with a GTR + G + I model, followed by extraction of phylogenetic pairwise distances based on branch lengths among *Harpagifer* individuals, using the APE package (v5.6-2) in R (v4.1.2). For co-phylogeny analysis, the phylogenetic tree of the haplotype sequences was reconstructed using PhyML algorithm, with a GTR + I model. Both phylogenetic trees of *Harpagifer* individual and haplotype sequences were reconstructed using NGPhylogeny (47), with substitution models chosen based on the implemented Smart Model Selection tool (SMS) and complemented by non-parametric SH-like branch supports, and were rooted using the midpoint rooting method (48) to avoid long-branch attraction.

## Statistic analyses for phylosymbiosis detection

The bacterial OTU table was rarefied at 5,750 sequences and converted into Bray-Curtis and weighted UniFrac dissimilarity distance matrices. To test the effect of host species identity on *Harpagifer* GMM composition, a permutational multivariate analysis of variance (PERMANOVA) was performed using Vegan (v2.6-2) and pairwiseAdonis (v0.4) R packages.

Due to substantial geographical distances among sites and the challenging access to these remote sites, sampling at the same time was unfeasible (Table 1). Therefore, a set of 14 environmental abiotic variables presumed important in influencing microbiome structure was extracted for each sampling locality from the Bio-ORACLE database (Fig. S1). This database offers dual benefits by covering all the sampling sites and providing averaged data over an extended period (i.e., 2000–2014), allowing us to mitigate the

unavoidable seasonal and annual variability of marine environmental properties (49). Additionally, to address potential intra-site depth variability during the sampling of *Harpagifer* individuals, the variables were extracted at the mean depth of each site. All the environmental variables, standardized to a mean of zero and a standard deviation of one, were analyzed using a principal component analysis. The scores of the samples on the first two principal components (capturing 91% the variability) were transformed into Euclidean distance using the vegdist function of the R package Vegan (v2.6-2) and used as environmental distance matrix (Fig. S1).

The geographic distances among sampling localities were obtained by converting longitude and latitude coordinates into kilometers with the earth.dist function implemented in the Fossil package (50), followed by a transformation with the Hellinger method using the decostand function of the Vegan package in R.

All matrices were standardized using the scale function implemented in R. The correlation between the dissimilarity distance matrix of the *Harpagifer* GMM with each one of the three explicative distance matrices (i.e., environmental, geographic, and host phylogeny) was examined with Mantel and partial Mantel tests implemented in Vegan, using Pearson correlation and 9,999 permutations (51). Furthermore, the respective contribution to GMM composition of these explanatory matrices was inferred using the distance-based multiple matrix regression with randomization (MMRR) approach (52), implemented in the R package PopGenReport (v3.0.7).

## Core microbiome definition and neutral model fitting

To identify bacterial taxa that are common to the eight *Harpagifer* populations, a core microbiome was defined at the OTU level, based on a minimum prevalence criterion >40% across all gut mucosa samples from our data set, as used in previous studies (18). The relationship between the prevalence and abundance of core microbiome OTUs was compared to the neutral community model proposed by Sloan et al. (53) and formalized in R by Burns et al. (54). Well-predicted OTUs (i.e., with abundance and prevalence comprised within the 95% confidence limits of the model) are supposed to be driven by stochastic factors (e.g., dispersal and ecological drift), while OTUs that deviated from the 95% confidence interval (CI) of the neutral model predictions are more likely influenced by deterministic factors (e.g., host selection).

## Microdiversity analysis and co-phylogeny testing

The microdiversity of each core OTU ($n$ = 17) was resolved into oligotypes through the minimum entropy decomposition (MED) algorithm developed by Eren et al. (39). Briefly, MED allows discrimination of the biologically meaningful microdiversity contained within one OTU from the stochastic noise caused by random sequencing errors. The oligotypes' phylogenetic tree of each core OTU from the core microbiome was inferred using PhyML algorithm in NGPhylogeny (47), with the default parameters and model selection determined by the implemented SMS. Co-phylogeny patterns between the bacterial oligotypes and *Harpagifer* species phylogenetic trees were tested through a total of 10 runs with 999 permutations of the Parafit function, implemented in the Ape package (v5.6-2). The Parafit function returned the relative contribution of each individual host-oligotype link to the co-phylogenetic model, with their associated *P*-values adjusted using the Benjamini-Hochberg procedure. For the graphical representation of the co-phylogeny pattern for the most abundant core OTU (OTU2), both host and symbiont trees were transformed into ultrametric trees, links with *P*-values >0.055 were pruned, and the tanglegram was edited using the R package Phytools (v1.0-3) (55). To further investigate the mechanism behind the co-phylogenetic pattern of OTU2, we used the Procrustean Approach to Co-phylogeny (PACo) (56) implemented in R (57) to assess the interdependence of one phylogeny on the other. Four randomization algorithms were tested in PACo—"r0," "c0," "quasiswap," and "r" models. The "r0" model (referred as "symbiont") posits that symbionts track hosts' evolution, while the "c0" model (referred as "host") assumes that hosts track symbionts' evolution. The

"quasiswap" model (referred as "undetermined") does not infer direction in the tracking, and the "r2" model (referred as "specialist/generalist symbiont") posits that symbionts track hosts' evolution, with the co-phylogenetic signal influenced by the specialist/generalist feature of the symbionts (i.e., number of partners). The best co-phylogenetic model was determined by comparing the phylogenetic congruences of each model obtained through 20,000 permutations using the pairwise permutation test implemented in the R package RCOMPANION (v2.4.16). The haplotype network of the OTU2 was reconstructed as previously described for the host (Host genetic diversity, genetic distance, phylogeographic structure, and phylogenetic reconstruction section). Distance-based redundancy analysis (db-RDA) implemented in the R package VEGAN was used to quantify the contribution of the *Harpagifer* species to the variations in OTU2 oligotype composition.

## RESULTS

### *Harpagifer* genetic diversity and phylogeographic structure

The COI diversity indexes for each analyzed *Harpagifer* species are summarized in Table S1. A total of 81 partial sequences of 664 nucleotides length were obtained, and 25 haplotypes were identified from the complete data set (Table S2). Low p-distance values were found among species, ranging from 0.29% between *H. antarcticus* and *H. georgianus* to 0.96% between *H. bispinis* and *H. kerguelensis* (Table S3), suggesting a recent diversification of these four species. Permutation tests of individuals between species showed highly significant differentiation among species, except for the comparison between *H. antarcticus* and *H. georgianus* ($P = 0.46$). Notably, the haplotype network reveals that *H. antarcticus* shared three haplotypes with *H. georgianus* (Fig. S2). Additionally, a single mutation distinguishes the *H. kerguelensis* haplogroup from the *H. antarcticus-georgianus* one, while a difference of three mutations distinguishes the *H. bispinis* haplogroup from the *H. antarcticus-georgianus* one, suggesting again the existence of three distinct groups corresponding to Patagonian populations (PAT1 and PAT2, hereafter called PAT), Kerguelen Islands population (KER), and Antarctic and South Georgia populations (WAP and SOG) (Fig. S2; Table S3).

### *Harpagifer* species identity influences its gut mucosa microbiome

Out of the 81 *Harpagifer* individuals sampled, 77 gut mucosa samples were successfully processed through metabarcoding, resulting in 2,990,285 cleaned sequences partitioned into 34,419 OTUs at 97% (Table 1). A weak but significant effect of *Harpagifer* species identity on GMM composition was detected through PERMANOVA using both Bray-Curtis (F-statistics = 4.46, $R^2 = 0.15$, $P < 0.001$) and weighted UniFrac metrics (F-statistics = 4.33, $R^2 = 0.15$, $P < 0.001$). All pairwise comparisons of GMM compositions among *Harpagifer* species were significantly different (pairwise PERMANOVA, $P < 0.04$) except between *H. antarcticus* (WAP) and *H. georgianus* (SOG) (Table S4), mirroring the absence of genetic differentiation among host haplotypes from these two regions (Fig. S2). The differences in microbiome composition between gut mucosa from KER and PAT were weak (pairwise PERMANOVA, $P = 0.04$), echoing the relatively more similar seawater properties of these two regions (Fig. S1).

### Significant phylosymbiosis of *Harpagifer* species and their gut mucosa microbiome

To gain further insights into the assembly mechanisms of the *Harpagifer* GMM, we tested whether the Bray-Curtis dissimilarity distance correlates with the geographic, environmental, and host phylogenetic distance matrices. The Mantel correlation tests revealed that environment, host phylogeny, and geography explained significant and relatively comparable amounts of variability in GMM of *Harpagifer* species (Table 2). When evaluating the independent effect of each explanatory matrix by controlling for the influence of other matrices variations with partial Mantel tests, both the host

**TABLE 2** Mantel test analysis on GMM of *Harpagifer* spp., using Bray-Curtis distances[a]

| Factors | Statistical test | $R^2$ | P-value |
|---|---|---|---|
| Environmental distance | Mantel | 0.33 | **<0.001** |
| Geographic distance | Mantel | 0.22 | **<0.001** |
| Host phylogeny | Mantel | 0.31 | **<0.001** |
| Environmental \| Geographic | Partial Mantel | 0.27 | **<0.001** |
| Environmental \| Host phylogeny | Partial Mantel | 0.15 | **<0.001** |
| Geographic \| Environmental | Partial Mantel | 0.10 | **<0.001** |
| Geographic \| Host phylogeny | Partial Mantel | 0.14 | **<0.001** |
| Host phylogeny \| Environmental | Partial Mantel | 0.08 | **<0.001** |
| Host phylogeny \| Geographic | Partial Mantel | 0.27 | **<0.001** |

[a]A total of 10,000 permutations were performed. *P*-values in boldface are considered as significant (α = 0.05). For partial Mantel, the vertical bar "|" means "controlling for."

phylogeny and the environment (and to a lower extent the geography) still significantly correlated with beta-diversity variations of *Harpagifer* GMM, even though with reduced explanatory power (Table 2). When dissimilarities of *Harpagifer* GMM were calculated using the weighted UniFrac metric, the degrees of correlation were lower and homogeneous across the explanatory matrices, remaining statistically significant (Table S5).

When examined through the MMRR analysis, the relative contribution of each matrix on bacterial composition of *Harpagifer* GMM remained consistent with the previous Mantel tests, with the highest contribution for the environment (coefficient of 0.14, $P < 0.001$), followed by the host phylogeny (coefficient of 0.10, $P < 0.001$) and geography (coefficient of 0.08, $P < 0.001$). The Mantel test performed with the combined distance

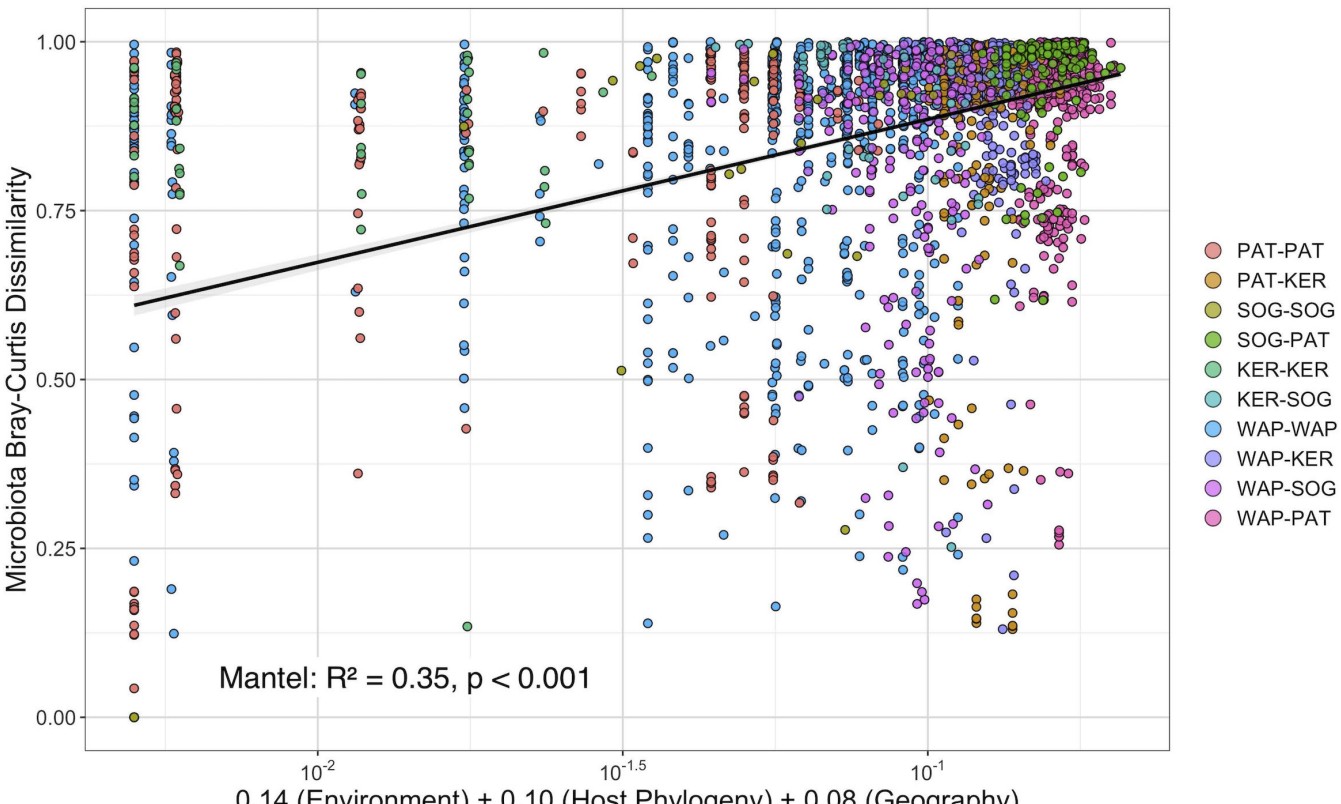

**FIG 2** Scatter plots showing the relationship between the Bray-Curtis dissimilarity of *Harpagifer* mucosa microbiome and the joint effect of host phylogenetic, geographic, and marine environmental distances based on the results of a multiple matrix regression with randomization analysis. Color of the points represents pairwise comparisons among sampling localities.

matrix computed from the three original ones weighted by their respective MMRR coefficient provided the best-fit model ($R^2 = 0.35$, $P < 0.001$) (Fig. 2).

## Reduced core microbiome in *Harpagifer* spp. gut mucosa

A reduced core microbiome comprising 17 OTUs was detected across all *Harpagifer* species studied (Fig. 3). Constituting less than 0.2% of the total OTU richness in GMM, this core microbiome exhibited a relative abundance of 22.5% ± 2.9% in gut mucosa samples, encompassing representatives from nine bacterial classes, predominantly within the Gammaproteobacteria phylum (Table 3). Prevalence of core OTUs ranged from 40% up to 96% of the gut mucosa samples. Two core OTUs were particularly abundant (OTU2 and OTU4, representing 12.6% and 6.9% of the total relative abundance in the whole GMM data set, respectively), while the 15 others had relative abundance <0.9% (Table 3).

The fitting of GMM composition to the neutral model was relatively low, explaining no more than 20% of the gut microbiome variance of *Harpagifer* ($m = 0.004$, $R^2 = 0.20$, Fig. 4). Remarkably, >88% of the core OTUs had an abundance-prevalence relationship that deviated from the neutral model predictions; most of them being either over- or under-represented (Fig. 4; Table 3).

The most abundant and prevalent OTU in the core microbiome of *Harpagifer*, namely, OTU2 affiliated with *Aliivibrio*, exhibited a lower frequency (and higher abundance) than the 95% CI predicted by the neutral model, suggesting that this OTU is either selected against by the host (i.e., invasive pathogenic taxa) or experience significant dispersal limitations (i.e., low probability of successful host colonization).

A total of 393,084 sequences, constituting 51% of the GMM, were affiliated with the *Aliivibrio* genus. Notably, OTU2 accounted for 99.4% of these sequences and, consequently, is referred to hereafter as *Aliivibrio*. The closest sequence retrieved from Blast

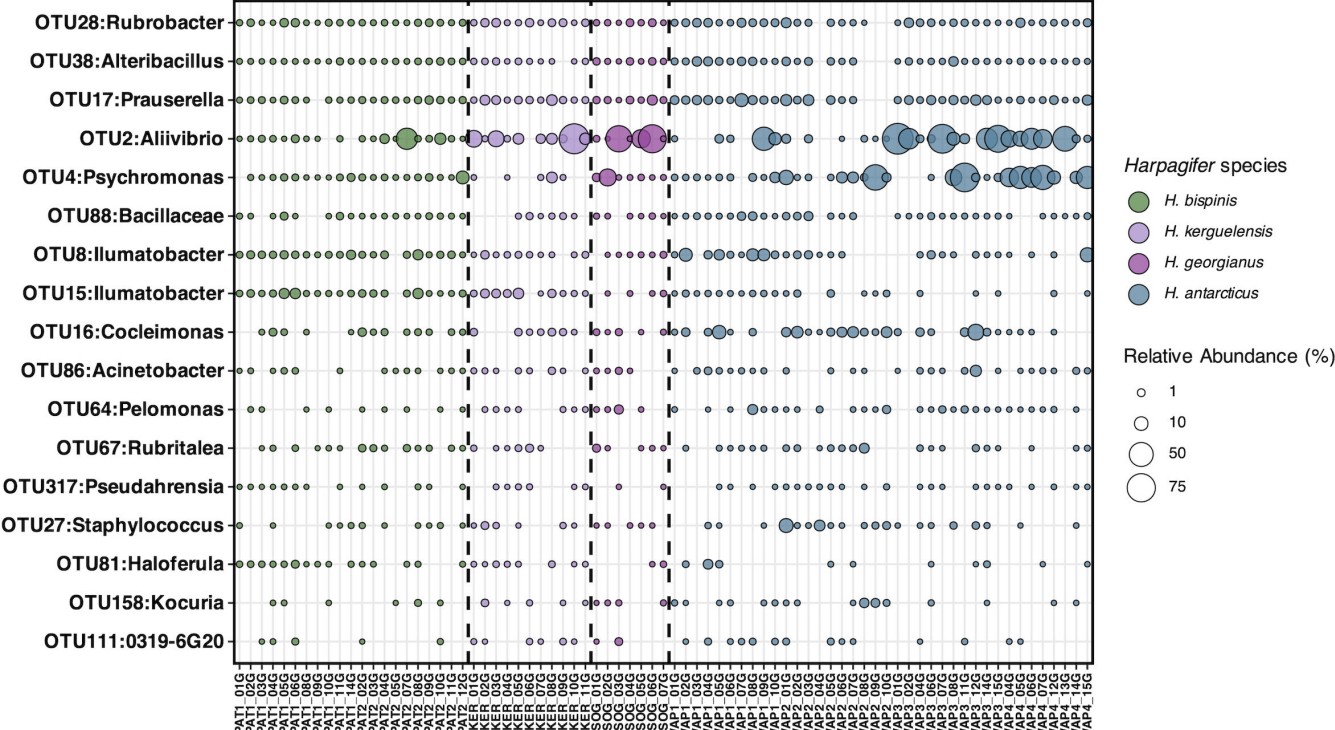

**FIG 3** Bubble plot of core microbiome mucosa from *Harpagifer* sp. gut mucosa defined at >40% prevalence across samples. Relative abundances are presented per sample, with distinct colors assigned to different *Harpagifer* species, visually demarcated by dashed lines. The taxonomic affiliations of the core OTUs are provided at the genus level, except for OTU88, which is identified at the family level. Specific relative abundance values of the core OTUs can be found in Table 3. Sample names are provided in x-axis, with the first four characters corresponding to the sampling site designation (see Table 1), and followed by the number of the *Harpagifer* individual.

TABLE 3  Fit of the core microbiome taxa from *Harpagifer* sp. gut mucosa to the co-phylogenetic model at the microdiversity level[a]

| OTU | Class | Final affiliation | Prev. | Rel. Abund. | Neutral model | Oligotype richness | ParaFit statistic | ParaFit P-value | Links WAP/SOG | Links KER | Links PAT |
|-----|-------|-------------------|-------|-------------|---------------|--------------------|-------------------|-----------------|---------------|-----------|-----------|
| OTU28 | Rubrobacteria | Rubrobacter | 96.1 | 0.20 | Over | 71 | ns[b] | ns | – | – | – |
| OTU38 | Bacilli | Alteribacillus | 93.4 | 0.34 | Over | 121 | ns | ns | – | – | – |
| OTU17 | Actinobacteria | Prauserella | 93.4 | 0.86 | Over | 63 | ns | ns | – | – | – |
| OTU4 | Gammaproteobacteria | Psychromonas | 82.9 | 6.90 | Under | 84 | ns | ns | – | – | – |
| **OTU2[d]** | **Gammaproteobacteria** | **Aliivibrio** | **81.6** | **12.60** | **Under** | **109** | **0.066 ± 0** | **0.001 ± 0.000** | **56.0** | **16.2** | **27.8** |
| OTU88 | Bacilli | Bacillaceae | 80.3 | 0.15 | Over | 38 | ns | ns | – | – | – |
| OTU8 | Acidimicrobiia | Ilumatobacter | 80.3 | 0.77 | Fitted | 109 | 0.147 ± 0 | 0.001 ± 0.000 | 1.1 | 1.8 | 97.1 |
| OTU15 | Acidimicrobiia | Ilumatobacter | 71.0 | 0.32 | Over | 130 | ns | ns | – | – | – |
| OTU16 | Gammaproteobacteria | Cocleimonas | 67.1 | 0.69 | Fitted | 243 | 1.321 ± 0 | 0.001 ± 0.000 | 65.7 | 0.0 | 34.3 |
| OTU86 | Gammaproteobacteria | Acinetobacter | 64.5 | 0.12 | Over | 69 | 0.007 ± 0 | 0.011 ± 0.001 | 54.2 | 16.7 | 29.2 |
| OTU64 | Gammaproteobacteria | Pelomonas | 60.5 | 0.10 | Over | 31 | ns | ns | – | – | – |
| OTU67 | Verrucomicrobiae | Rubritalea | 60.5 | 0.13 | Over | 104 | 0.580 ± 0 | 0.001 ± 0.000 | 51.9 | 2.8 | 45.3 |
| OTU317 | Alphaproteobacteria | Pseudahrensia | 56.6 | 0.01 | Over | 18 | ns | ns | – | – | – |
| OTU27 | Bacilli | Staphylococcus | 53.9 | 0.18 | Over | 90 | 0.015 ± 0 | 0.004 ± 0.001 | 22.2 | 11.1 | 66.7 |
| OTU81 | Verrucomicrobiae | Haloferula | 46.1 | 0.11 | Over | 94 | 0.033 ± 0 | 0.001 ± 0.001 | 55.4 | 1.2 | 44.6 |
| OTU111 | Oligoflexia | 0319-6G20 | 40.8 | 0.08 | Over | 32 | ns | ns | – | – | – |
| OTU158 | Actinobacteria | Kokuria | 40.3 | 0.08 | Over | 33 | 0.003 ± 0 | 0.046 ± 0.002 | 72.7 | 0.0 | 27.3 |

[a]Prev., OTU prevalence (%) among all samples of the data set; Rel. Abund., relative abundance (%) of the OTU within the gut mucosa data set. Mean values with standard errors are presented for the global ParaFit statistics and *P*-values. Links WAP/SOG, KER, and PAT (%), percentage of significant links involving Harpagifer haplotypes from West Antarctic Peninsula and South Georgia, Kerguelen Islands, and Patagonia, respectively.
[b]ns, non-significant *p*-value.
[c]–, Undetermined.
[d]Selected OTU for further analysis and graphical representation.

analysis of the *Aliivibrio* representative sequence matched with an uncultured bacterium clone (99.3% identity), previously retrieved from the gut of *Notothenia coriiceps* (suborder Notothenioidei) fished in the Antarctic Peninsula (23).

## Co-phylogeny between *Harpagifer* species and some core members of the gut mucosa microbiome

Eight out of the 17 core OTUs harbored significant signatures of co-phylogeny at the microdiversity level with *Harpagifer* spp. ($P < 0.05$ for each OTU) (Table 3), suggesting that the evolution of these gut mucosa OTUs and their hosts was not independent (rejection of the $H_0$ of Parafit test). The fit values were relatively low (ParaFit statistic <1.3), suggesting that external factors would also shape the observed patterns of co-phylogeny. Due to the absence of differences in GMM composition and host genetic divergence, WAP and SOG were combined in the results presentation of the host-oligotype links and co-phylogenetic representation. The proportion of significant links (i.e., associations between specific bacterial oligotypes and a given host species) involving WAP/SOG and PAT populations was globally similar, representing 47.4% ± 8.4% and 46.5% ± 8.6% of all significant links, respectively. Contrastingly, fewer links involving *Harpagifer* haplotypes from KER contributed to the global fit of the co-phylogeny model (8.3% ± 2.6%) (Table 3).

Due to its high abundance and prevalence in the GMM (deviating from neutral model predictions) and its significant co-phylogeny signal, a special interest was given to the *Aliivibrio* genus. The db-RDA analysis confirmed a significant influence of the host species identity on *Aliivibrio* oligotype composition ($R^2 = 0.07$, $P < 0.01$, Fig. 5A). Pairwise PERMANOVA comparisons showed that *Aliivibrio* oligotype composition was only different between *H. antarcticus* from WAP and *H. bispinis* from PAT ($R^2 = 0.05$, $P = 0.001$). The oligotype network of *Aliivibrio* graphically confirmed that most of the predominant oligotypes were principally shared among SOG, WAP, and KER regions, while the PAT oligotypes tended to be more exclusive to this region (Fig. 5B).

Beyond the qualitative effect of the host species identity, the phylogeny of *Aliivibrio* mirrors the phylogenetic patterns in *Harpagifer* (Fig. 5C). The significant links involved

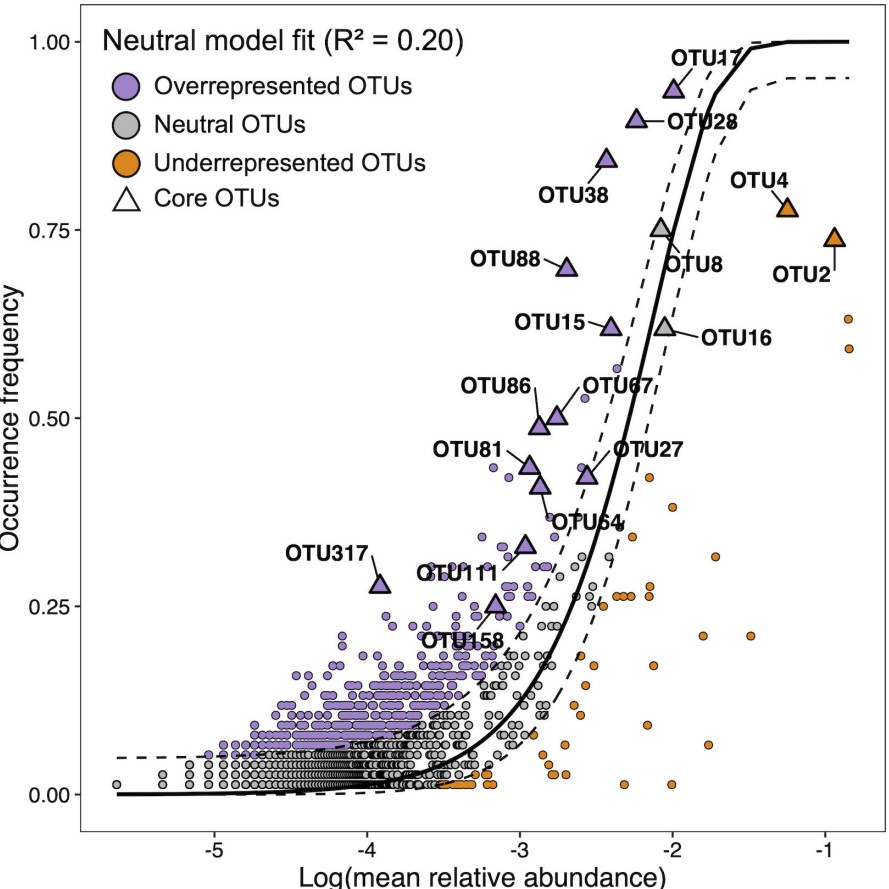

**FIG 4** Neutral model fit for the gut mucosa microbiome (solid dark line) with bootstrap 95% CI (dashed dark line). OTUs depicted in gray have a frequency of occurrence in the metacommunity congruent to their abundance under the neutral model hypothesis. OTUs appearing above the 95% CI (purple) or below the 95% CI (orange) are significantly more frequent (overrepresented) or less frequent (underrepresented), respectively, than predicted by the model in the metacommunity of *Harpagifer* gut mucosa microbiome. Triangles designate the OTUs that were identified as part of the gut mucosa core microbiome.

22 host haplotypes (out of the 25) and 51 *Aliivibrio* oligotypes (out of 60). The *Aliivibrio* oligotypes conformed two distinct clades, potentially representing below-genus divergence at the 16S V3–V4 locus. The *Aliivibrio* oligotypes significantly associated with *H. bispinis* clustered separately from the oligotypes significantly associated with all the other *Harpagifer* species, while *Aliivibrio* oligotypes significantly associated with *H. kerguelensis*, *H. antarcticus*, and *H. georgianus* clustered together, consistently with the host haplotypes clustering (Fig. 5C). PACo analysis of *Aliivibrio* oligotypes revealed that the "r2" model led to the highest phylogenetic congruence between host and microbe phylogenies, suggesting that the *Aliivibrio* phylogeny is driven by *Harpagifer* phylogeny (and not the opposite) and that the degree of specialization of *Aliivibrio* oligotypes (quantified by the number of associations with *Harpagifer* haplotypes) also contributed to the global fit of the co-phylogeny model (Fig. S3).

## DISCUSSION

In this study, we investigated how the evolutionary changes among *Harpagifer* closely related species associate with structural changes in their GMM. To the best of our knowledge, this study represents the first large-scale characterization of the GMM of a fish genus inhabiting the SO. In natural systems, environment, geography, and host phylogeny are generally confounded in the microbiota assembling among different

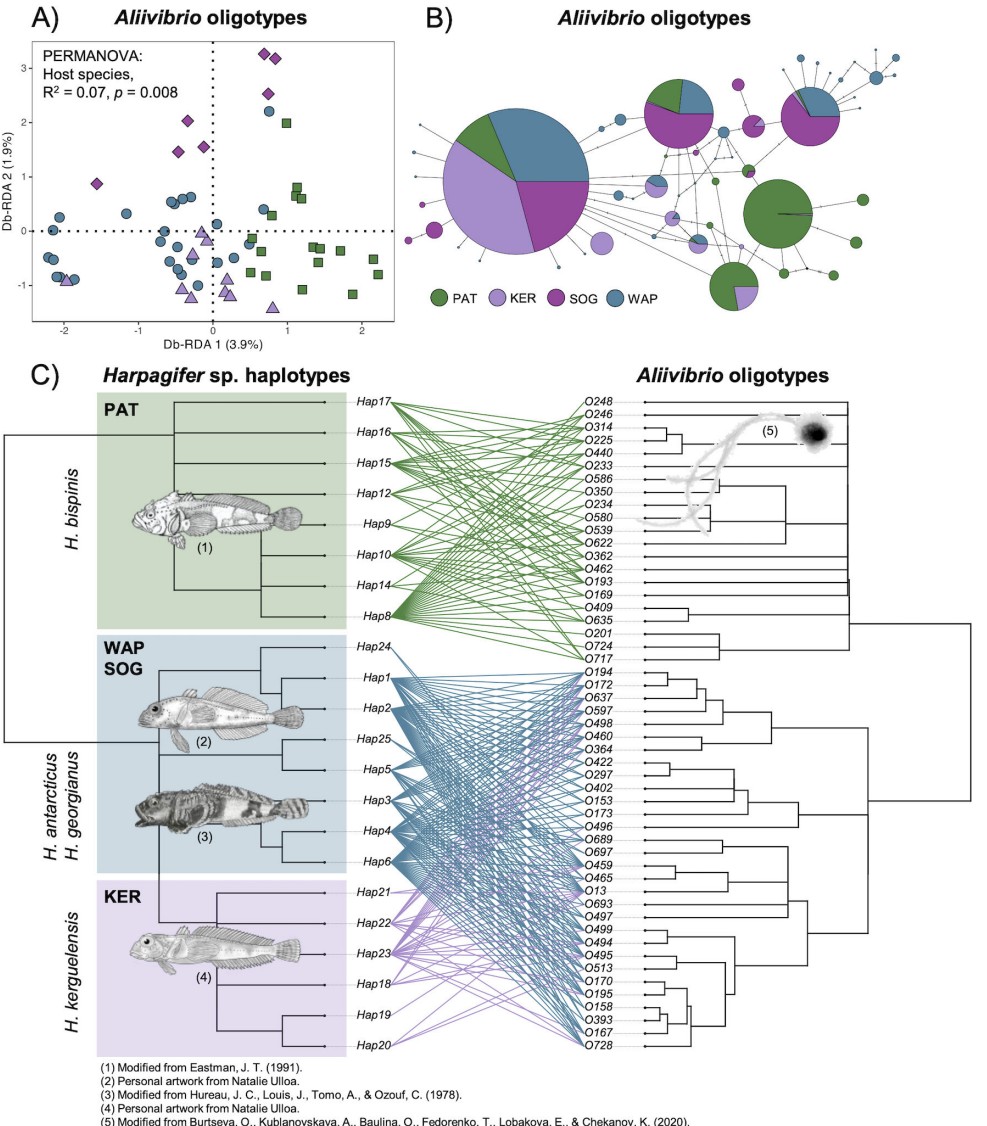

(1) Modified from Eastman, J. T. (1991).
(2) Personal artwork from Natalie Ulloa.
(3) Modified from Hureau, J. C., Louis, J., Tomo, A., & Ozouf, C. (1978).
(4) Personal artwork from Natalie Ulloa.
(5) Modified from Burtseva, O., Kublanovskaya, A., Baulina, O., Fedorenko, T., Lobakova, E., & Chekanov, K. (2020).

**FIG 5** Example of co-phylogenetic pattern among *Aliivibrio* and *Harpagifer* spp. (A) Distance-based redundancy analysis (db-RDA) quantifying the contribution of the host species to explain the variations in *Aliivibrio* oligotypes composition in *Harpagifer* sp. gut microbiome. (B) Median joining network of *Aliivibrio* oligotypes. Colors are assigned to the host biogeographic regions. The size of the circles is scaled on oligotypes' frequencies. (C) Pruned tanglegram of the co-phylogenetic relationships between 22 *Harpagifer* haplotypes and 51 *Aliivibrio* oligotypes. The *Harpagifer* and *Aliivibrio* trees are ultrametric. The significant links between *Harpagifer* haplotypes and *Aliivibrio* oligotypes were plotted in the figure (ParaFit, $P < 0.05$).

species (4, 10). However, controlling the environmental conditions to isolate the contribution of host genetics also distorts the "wild" microbiome, since the host must be maintained in captivity (58, 59). Moreover, the phylosymbiosis can emerge from various sources, including host evolutionary history, shared ecology linked to phylogenetically conversed host traits (e.g., diet, habitat, and host immune systems), and vertical transmission of microbiome; its detection is independent of inferring the underlying mechanisms and does not necessarily require overlapping distribution of the different species tested (16, 60). Therefore, combining a broad sampling strategy of *Harpagifer*, encompassing the three major biogeographic regions of its distribution area across the SO, and adapting statistical tools to explore the relative contributions of each factor (i.e., Mantel and MMRR) are valid approaches to provide a comprehensive evaluation of *Harpagifer* GMM assembly factors and to detect phylosymbiosis (4, 61).

Our findings indicate that *Harpagifer* host identity contributed to the variations observed in GMM composition, albeit to a limited extent. This result suggests that, despite their distribution in distinct biogeographic regions, different *Harpagifer* species still share similar ecology leading to relatively similar gut microbial communities, as expected in recent allopatric speciation scenario (6, 16) and in line with existing knowledge regarding the dietary consistency among these species (35, 36). Beyond the mere host species identity effect, we showed that the phylogenetic distances among *Harpagifer* host species substantially correlate with gut bacterial dissimilarity distances, thus fulfilling the two central tenets associated with phylosymbiosis signature (6, 10). In other terms, when conducting pairwise comparisons among *Harpagifer* species—except for the comparison between *H. georgianus* and *H. antarcticus*—the gut microbiomes show greater similarity among individuals of the same species compared to individuals of different species, and these similarities are congruent with the branching pattern of the species phylogeny. Studies investigating phylosymbiosis in the gut microbiome of anciently diverged wild marine fish are sparse, and the reported cases vary from absent to moderate signal, with a limited understanding of the factors that influence its intensity (22, 62, 63). In agreement with our hypothesis, the recent allopatric diversification of the *Harpagifer* genus across the SO is associated with a clear phylosymbiosis signal, thus expanding to marine fish the statement that phylosymbiosis strength among vertebrates and their microbiomes depends on the age of the last common ancestor (16). Although phylosymbiosis alone could arise from different evolutionary mechanisms unrelated to co-diversification, such as host filtering (12), its detection provides a strong basis for further investigating co-phylogeny of specific bacterial taxa shared among *Harpagifer* species (64).

Descriptions of core gut microbiomes in different species of wild marine fishes are rare and have been exclusively achieved so far across sympatric species from the same diet category (65–67). Huang et al. (68) did not find any common taxa in the gut microbiome of 20 marine fish species from coastal waters of Hong Kong, due to the high dependency of the gut core microbiome on the host's feeding habits. Here, we revealed the existence of a core microbiome across gut mucosa of four *Harpagifer* species (77 individuals) distributed in three geographically distant regions of the SO. This core was characterized by a relatively low diversity compared to the total OTU richness in GMM and by the high dominance of a single taxon, as previously reported in fish core microbiomes (68–70). While the persistence over time of the core taxa remains undetermined, the observation that most of these OTUs deviated from neutral model predictions and were detected across large geographical distances suggests a major role of *Harpagifer* selective constraints in the recruitment, assembly, and maintenance of its gut microbiome (70).

Around half of the core bacterial taxa harbored strong signal of co-phylogeny with *Harpagifer* at microdiversity resolution. These taxa are expected to present a certain degree of host specificity and to share at least some part of the host evolutionary history (4, 71). This result constitutes an unprecedented step forward in the understanding of marine fish holobiont, considering that most of the studies so far barely detected phylosymbiosis. Interestingly, the most abundant and prevalent taxon of the core GMM exhibited a significant co-phylogenetic pattern. This taxon belonged to the *Aliivibrio* genus, which has been identified in the past as a major component of the gut microbiome of notothenioid fishes (23, 24), and frequently associated with other fish species such as the European seabass and Atlantic salmon (72, 73). While there is no direct evidence of its ecological role in the fish holobiont, some authors suggest that *Aliivibrio* would be commensals and able to readily colonize the fish intestines (24, 74) and form biofilm onto intestine mucosa surface (75, 76). This commensalism would be mediated by the capacity of *Aliivibrio* to degrade chitin, a highly conserved metabolism in the *Vibrionaceae* family, previously confirmed by the detection of chitinase activity in some *Aliivibrio* strains (77). Chitin is the most abundant biopolymer in the ocean (78), since it constitutes the exoskeleton of crustaceans such as copepods, amphipods, and krill

(79, 80). These crustaceans are dominant in the diet of *H. antarcticus*, *H. georgianus*, and *H. bispinis*, comprising between 70% and 100% of the biomass ingested (35, 36). Additionally, *Aliivibrio*, along with other together with other *Vibrionaceae*, are common members of crustacean microbiota, such as copepods (81). Furthermore, as the microbial metabolization of chitin aminopolysaccharide provides substantial source of carbon and nitrogen easily accessible for the host, a mutualistic cross-feeding is imaginable between *Harpagifer* and *Aliivibrio*, providing an ecological advantage to the holobiont (82, 83). Alternatively, other studies described the opportunistic and potentially pathogenic status of *Aliivibrio* strains inhabiting the intestinal tract of fishes (73, 74). The deviation of *Aliivibrio* (i.e., occurring less frequently than expected by the neutral model analysis) suggests a selection against it by the host, in line with a possible parasitic behavior of *Aliivibrio* (i.e., more abundant than expected) (54, 73).

Since several *Aliivibrio* oligotypes co-occurred within the same *Harpagifer* species and several *Harpagifer* species hosted the same *Aliivibrio* strain, we characterized the *Harpagifer-Aliivibrio* holobiont as a diffuse symbiosis (11). Unlike specialist symbionts that are highly specific to their hosts associations, more generalist symbionts generally show low phylogenetic congruence with their hosts (84). Accordingly, we found that the *Aliivibrio* oligotypes were less specific to the haplotypes of *H. kerguelensis* from KER, being frequently associated with *H. antarcticus* and *H. georgianus* haplotypes from WAP/SOG as evidenced by the links in the tanglegram representation. It is highly unlikely that this unspecific interactions' pattern resulted from the unbalanced sampling of *Harpagifer* individuals across biogeographic region, as consistent numbers of host haplotypes were identified (i.e., WAP/SOG: 9, KER: 6, PAT: 10). We rather suggest that repeated host switch events generate the imperfect match between *Aliivibrio* and *Harpagifer* phylogenies. Although the transmission mode of *Aliivibrio* symbionts remains unknown, the diffuse symbiose and low phylogenetic congruence patterns suggest a horizontal transmission, either potentially widespread and repeatedly acquired from the surrounding environment or vertically transmitted from parent to offspring (11), facilitating colonization of novel *Harpagifer* specimens and recurrent host-switch events. Although specific data on the relative abundance of planktonic *Aliivibrio* in the Southern Ocean are lacking, a plausible explanation for the widespread detection of *Aliivibrio* sp. in fishes is the frequent colonization of new individuals within a same fish species through the excretion of *Aliivibrio*-rich feces into the surroundings (74). Moreover, as other *Vibrionaceae*, *Aliivibrio* is predominantly found attached to zooplankton rather than existing in a free-living state in seawater (85), possibly explaining its facile colonization of *Harpagifer* gut.

A strong co-phylogeny signal was observed between *Harpagifer* and *Aliivibrio*, mostly driven by the Patagonian and Antarctic/South Georgian oligotypes of *Aliivibrio* conforming two distinct sub-clades predominantly associated with their respective *Harpagifer* host species. According to PACo analyses, the most likely co-phylogenetic model was the adaptive tracking of *Harpagifer* phylogeny by *Aliivibrio*, suggesting that the diversification of *Aliivibrio* would result from unidirectional selection toward *Harpagifer* rather than independent response to a same biogeographic event or co-evolution (5, 56). Taking together, these results indicate that the biogeographic event experienced by *Harpagifer* [i.e., intensification of the Antarctic Polar Front (APF) (34)] leading to the speciation of *H. bispinis* by vicariance indirectly generated the co-diversification of its specific *Aliivibrio* symbionts. An important prerequisite to this geographic model of co-diversification proposed by Groussin et al. (9) is the dispersal limitation of the microbial symbionts. The absence of significant links between *Aliivibrio* oligotypes from the PAT-related sub-clade with the *Harpagifer* haplotypes from WAP/SOG and KER and the phylogeographic structure observed in *Aliivibrio* oligotypes network support the relatively limited dispersal capacity between these two regions (54). Consistently, a previous study demonstrated that the APF limits the genetic connectivity among marine bacterial populations associated to invertebrate gut mucosa (i.e., sea urchin *Abatus*) from ANT and PAT (86). Contrastingly, some *Aliivibrio* oligotypes indifferently

associate with *H. kerguelensis* and *H. antarctica*, further emphasizing that unlike host specimens, bacterial populations from ANT and KER may be occasionally interconnected generating a more diffuse pattern of association and counteracting *Aliivibrio* vicariance (86, 87). Alternatively, given the potentially more recent divergence of *H. antarcticus*/*H. georgianus* and *H. kerguelensis* (p-distances <1.62), compared to *H. antarcticus* and *H. bispinis* (p-distance >4.2), it is plausible that the low mutation rate of the 16S rRNA gene hinders the complete discrimination of *Aliivibrio* strains associated with these three species (8). On the host side, the COI gene appears to be insufficient to fully discriminate *H. antarcticus* from *H. georgianus* species, probably due to the result of maintenance of ancestral polymorphism or incomplete lineage sorting (88). Future investigations will need to incorporate more resolutive genetic markers for both host and bacterial symbionts, along with the *Harpagifer* species not covered in our study, to fully resolve the co-phylogeny between the two partners. Finally, the synchronicity of *Harpagifer* and *Aliivibrio* divergence requires further exploration using a robust time-calibrated phylogeny of the symbiont to confirm the strict co-diversification of hosts and symbiont speciation times (9).

In conclusion and contrastingly to the previously studied fish models, we revealed that host phylogeny was a substantial predictor of GMM composition of *Harpagifer*. Our survey represents the most conclusive evidence to date that phylosymbiosis and co-phylogeny occur between teleost fishes and their microbiome across the SO and that recently diverged closely related species are suitable models to unravel phylogenetic congruency signals. We identified a small subset of bacterial taxa harboring robust co-phylogeny signal, largely dominated by *Aliivibrio*. These taxa are good candidates for further genomic-based exploration of their metabolic and ecological roles, due to their supposed tight and/or long-term interdependence with *Harpagifer*. While the co-diversification of *Harpagifer* and *Aliivibrio* remains to be confirmed, we provide a foundation to explore the mechanisms behind co-phylogeny signatures, notably by understanding the contribution of host biogeography into the diversification process of its symbionts.

## ACKNOWLEDGMENTS

This work was financially supported by the ANID-Millennium Science Initiative Program, ICN2021_002. The authors thank the French Polar Institute project 1044 PROTEKER for field access and the University of Wisconsin Biotechnology Center DNA Sequencing Facility (Research Resource Identifier, RRID:SCR_017759) for providing MiSeq sequencing facilities and services. This research was partially supported by the supercomputing infrastructure of the NLHPC (CCSS210001). We would like to express our gratitude to Zambra Lopez, Francisco Bahamondes, and Javier Cárcamo for their contributions in field sampling. We also thank Sebastián Rosenfeld and Natalie Ulloa for their graphical contributions to this study.

This study was supported by the ANID FONDECYT Postdoctoral Project (grant no. 3200036), ANID FONDECYT Regular Projects (grants no. 1151336 and 1211672), and ANID-Millennium Science Initiative Program (ICN2021_002).

E.P., J.O., L.C., and G.S. conceived and developed the study. G.S. secured the research funds. E.P., K.G., T.S., L.C., and G.S. contributed to the field work. G.S. performed the laboratory work and analyzed the molecular data. G.S. generated and analyzed the figures. G.S. drafted the manuscript, and all authors revised it and approved its final version.

## AUTHOR AFFILIATIONS

[1]Millennium Institute Biodiversity of Antarctic and Subantarctic Ecosystems (BASE), Santiago, Chile
[2]Department of Ecological Sciences, Faculty of Sciences, University of Chile, Santiago, Chile
[3]Institute of Ecology and Biodiversity, Santiago, Chile

[4]Aix Marseille University, Univ Toulon, CNRS, IRD, Mediterranean Institute of Oceanography (MIO) UM 110, Marseille, France, Marseille, France

[5]UMR 6282 Biogeosciences, University Bourgogne Franche-Comté, CNRS, EPHE, Dijon, France

[6]Laboratory of Antarctic and Subantarctic Marine Ecosystems, Faculty of Sciences, University of Magallanes, Punta Arenas, Chile

[7]Cape Horn International Center, Puerto Williams, Chile

## AUTHOR ORCIDs

Guillaume Schwob  http://orcid.org/0000-0003-0996-4060
Léa Cabrol  http://orcid.org/0000-0003-0417-2021

## FUNDING

| Funder | Grant(s) | Author(s) |
|---|---|---|
| ANID FONDECYT Postdoctoral Project | 3200036 | Guillaume Schwob |
| ANID FONDECYT Regular Project | 1211672 | Guillaume Schwob |
| | | Léa Cabrol |
| | | Julieta Orlando |
| | | Elie Poulin |
| ANID-Millenium Science Initiative Program | ICN2021_002 | Guillaume Schwob |
| | | Léa Cabrol |
| | | Julieta Orlando |
| | | Karin Gérard |
| | | Elie Poulin |

## AUTHOR CONTRIBUTIONS

Guillaume Schwob, Conceptualization, Data curation, Formal analysis, Funding acquisition, Investigation, Methodology, Project administration, Visualization, Writing – original draft, Writing – review and editing | Léa Cabrol, Conceptualization, Investigation, Validation, Writing – review and editing | Thomas Saucède, Investigation, Validation, Writing – review and editing | Karin Gérard, Investigation, Validation, Writing – review and editing | Elie Poulin, Conceptualization, Investigation, Supervision, Validation, Writing – review and editing | Julieta Orlando, Conceptualization, Investigation, Supervision, Validation, Writing – review and editing

## DATA AVAILABILITY

Amplicon sequences of 16S rRNA and COI have been deposited in the National Center for Biotechnology Information (NCBI), under the Sequence Read Archive (SRA) PRJNA803378, and in GenBank under the accession numbers ON891147 to ON891171, respectively.

## ETHICS APPROVAL

Animal management was approved by the Universidad de Chile Institutional Animal Care and Use Committee (resolution no. 20363-FCS-UCH).

## ADDITIONAL FILES

The following material is available online.

## Supplemental Material

**Supplemental material (Spectrum03830-23-s0001.docx).** Figures S1 to S3 and Tables S1 to S5.

## Open Peer Review

**PEER REVIEW HISTORY (review-history.pdf).** An accounting of the reviewer comments and feedback.

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
