## [Reviewer comments · Microbiology Spectrum]

Microbiology Spectrum

Unveiling the co-phylogeny signal between plunderfish *Harpagifer* spp. and their gut microbiomes across the Southern Ocean

Guillaume Schwob, Léa Cabrol, Thomas Saucède, Karin Gérard, Elie Poulin, and Julieta Orlando

Corresponding Author(s): Guillaume Schwob, Millennium Institute Biodiversity of Antarctic and Subantarctic Ecosystems

Review Timeline:

Submission Date:	November 1, 2023
Editorial Decision:	January 2, 2024
Revision Received:	January 25, 2024
Accepted:	February 9, 2024

Editor: Konstantinos Kormas

Reviewer(s): Disclosure of reviewer identity is with reference to reviewer comments included in decision letter(s). The following individuals involved in review of your submission have agreed to reveal their identity: Omar Mejia (Reviewer #1); Isabelle George (Reviewer #2)

Transaction Report:

DOI: <https://doi.org/10.1128/spectrum.03830-23>

Re: Spectrum03830-23 (Unveiling the co-phylogeny signal between plunderfish *Harpagifer* spp. and their gut microbiomes across the Southern Ocean)

Dear Dr. Guillaume Schwob:

Thank you for the privilege of reviewing your work. Below you will find my comments, instructions from the Spectrum editorial office, and the reviewer comments.

Please return the manuscript along with your point-by-point rebuttal within 60 days; if you cannot complete the modification within this time period, please contact me. If you do not wish to modify the manuscript and prefer to submit it to another journal, notify me immediately so that the manuscript may be formally withdrawn from consideration by Spectrum.

Revision Guidelines

Sincerely,
Konstantinos Kormas
Editor
Microbiology Spectrum

Reviewer #1 (Comments for the Author):

In this paper, the authors evaluate the possible cophylogenetic signal between a group of species of the genus *Harpagifer* and their gut mucosa microbiome. The paper represents a novel and a good contribution to the knowledge of the processes that drive the coevolution of host and associate microbiomes. The molecular techniques used are adequate as well as the analysis used.

Despite this, I find several sections where further details of the methods used need to be explained in detail, as well as some of the inferences and conclusions that are presented that require moderation in light of the flaws derived from an incomplete sample scheme of all Harpagifer species. I am including a series of recommendations that would improve the author's manuscript.

Major comments

In lanes 121 to 123 the authors state that "The diversification of the Harpagifer genus occurred 1.2-0.8 Myr ago, from Antarctica towards the Patagonia and sub-Antarctic areas during the Pleistocene." The family Harpagiferidae comprise one genus and 12 species included in the genus Harpagifer. According to Matschiner the diversification of the more derived notothenioid families including Harpagiferidae begin in the mid Miocene (HPD 9.9-20 Ma). The authors suggest a recent divergence of their studied species *H. antarcticus* and *H. bispinis* based on the calibration of Hüne et al. who only include these two species. The authors include in their analysis other two species, *H. georgianus* that cannot be discriminated from *H. antarcticus* and *H. kerguelensis*. In the absence of a phylogeny for the complete 12 species of the genus, it is impossible to state if *H. kerguelensis* is, in reality, the sister clade of the *H. antarcticus* + *H. bispinis* clade, I agree with that, but the authors cannot suggest a recent divergence among the four studied species unless they build a phylogenetic calibrated tree that includes also *H. kerguelensis*.

The authors report the collection of 77 individuals in Table 1, but in lane 239 they report that a total of 81 COI sequences were obtained, thus, the number of individuals and sequences does not match. On the other hand, they mention that a total of 25 haplotypes were recovered and formed three haplogroups of supplementary figure 2, but, the number of haplotypes in their figure 2 only comprise 22 haplotypes. Regarding the haplotype network, the authors suggest the recognition of three haplogroups, PAT, KER, and WAP+SOG. At first sight, their appreciation appears correct, however, Hap 13 (PT) is separated from Hap 5(WAP+SOG) by three mutation steps, but Hap5 (WAP+SOG) is separated by only one mutation step from Hap22(KER), on the other hand, Hap12 (PAT) is separated by four mutations from Hap8(PAT). Due to the low divergence of the COI sequences and the apparent maintenance of ancestral polymorphism, it is clear that a phylogenetic approach is not the best choice to support the number of haplogroups, but neither is an arbitrary separation based on a visual approach to the haplotype network. I recommend the authors the use of an approach to defining genetic clusters at the population level such as baps algorithm implemented in fastbaps library.

The authors report an ML phylogenetic tree obtained with PhyML, they need to report which substitution model was used and how it was chosen, and whether they performed branch support? Please explain. Additionally, their phylogenetic tree presents many polytomies in opposition to the phylogenetic tree of *Allivibrio* oligotypes. The V3 and V4 primers of Klindworth allow to amplify an amplicon of 464 bp, if the authors used a 97% similarity threshold to discriminate among OTUs it means that all the oligotypes of *Allivibrio* must be concentrated in a range of a maximum of 14 bp, it surprises me how a total of 51 oligotypes could be recognized within the genus *Allivibrio* and lead to an almost dichotomic tree in opposition to the phylogenetic tree of *Harpagifer* using the same methodological approach (ML) that as mentioned earlier need to be explained at the detail (substitution model used and how it was chosen).

The author aims to test the hypothesis of a co-phylogenetic signal between *Harpagifer* species and shared members of their gut microbiomes (i.e. core microbial taxa) (lanes 137 and 138), earlier in the introduction the authors state that cophylogenetic signal is validated by congruent topologies of host species and symbiotic phylogenies, I completely agree with their statement, however, this strong postulate is at the same time the major flaw of their paper. A phylogenetic tree allows to recover of genealogical relationships between ancestors and descendants, but, populations evolve in a different way as species. For example, the authors search to state cophylogenetic signal between haplotypes and microbiome, in this way, they have three haplotypes shared between two species, *H. antarcticus*, and *H. georgianus*, the fact that both species shared the same haplotype could be the result of maintenance of ancestral polymorphism or incomplete lineage sorting among others factors. Despite sharing the same COI sequence both species have different evolutionary histories, if the authors want to track the coevolutionary trajectories of host and associates they will need the complete monophyletic group of 12 species of the genus *Harpagifer*, I know that this escape from the main goal and I do not suggest to make it, but I suggest that the authors need to perform the cophylogenetic tests for each one of the four species and not for the complete dataset, at the same time I suggest the authors to "ease up" some of the sentences of their introduction due to their impossibility to include all the species included in the genus *Harpagifer* or at least they warn of the situation and limitation of their findings and conclusions.

The Mantel test was performed among GMM against host phylogeny, geography, and environmental distance matrices. The authors report a phylogenetic tree based on haplotypes but I suspect based on the results of the supplementary material 2 that the distance matrix for Mantel comparison of host phylogeny was performed only for the four species of the genus *Harpagifer*. It is not clear how this analysis was performed, if the authors used the four terminal species they need to explain how they estimated the GMM for each one of the *Harpagifer* species, mainly that they used Bray Curtis distance that is based on abundances, I assume that they had a GMM for each individual, for example, in *H. antarcticus* they have 38 individuals, how they obtain an average GMM for this species? On the other hand, if *H. antarcticus* and *H. georgianus* share the same haplotypes the genetic distance between this species must be zero, unless they use as genetic distance the private haplotypes for each species (HAP 24) for *H. georgianus* and HAP4, 6 or 7 for *H. antarcticus*, I am not able to understand how the authors perform this analysis, please explain at detail.

In lanes 332 to 338 the authors state "PACo analysis of *Aliivibrio* oligotypes (OTU2) revealed that the 'r2' model led to the highest phylogenetic congruence between host and microbe phylogenies, suggesting that the *Aliivibrio* phylogeny is driven by *Harpagifer* phylogeny (and not the opposite), and the degree of specialization of *Aliivibrio* oligotypes (quantified by the number of associations with *Harpagifer* haplotypes) also contributed to the global fit of the cophylogeny model " I agree with the fact that there is a clade of *Aliivibrio* oligotypes that extend from O248 to O717 that are exclusive of *Harpagifer bispinus* , nevertheless, these set of oligotypes form a polytomy at the base of the cladogram with the rest of the clusters present in the other *Harpagifer* species, so, the first split between *Allivibrio* oligotypes cannot be linked to the split between *Harpagifer* species, please change the redaction.

In the discussion lanes 361 to 363 the authors state "the gut microbiomes were more similar among individuals of the same *Harpagifer* species than among different species", however, in their results, they state in lanes 249 to 252 that "GMM comparisons were different except between *H. antarcticus* and *H. georgianus* mirroring the absence of genetic differentiation among these species", meanwhile, in lanes 253 and 254 they suggest that the weak differences in PERMANOVA reflect the similar seawater properties of both regions. I am confused, in the abstract the authors state that the core microbiome of *Harpagifer* is characterized by a low diversity mainly driven by selective processes, I assume that if a phyllosymbiosis signal is present the selective pressures are imposed by the host as suggested in lanes 389 to 393 and not by the environmental water parameters. I suggest the authors carefully review these sentences that appear contradictory.

Why to use the r2 option in Paco? Procrustes analysis of cophylogeny allows performing the analysis through different randomization approaches including r2 model. In their discussion, the authors state that " the most likely co-phylogenetic model was the adaptive tracking of *Harpagifer* phylogeny by *Allivibrio*..." Did the authors perform the PACo analysis using all randomization options? How do they compare them and choose the r2 as the best predictor? Their selection of r2 model must be fully justified in the methods.

In lanes 466 to 469 the authors state" the absence of genetic and phylogeographic structure between SOG and WAP question the validity of *H. antarcticus* and *H. georgianus*". I am not a specialist in the group and the information provided does not allow me to track the taxonomy of both species, but, Eschemeyer fish catalog recognizes both species as valid. Both, population genetics and phylogeography have their epistemology, but the analysis used by the authors does not correspond to either, moreover, as I mentioned several times, shared haplotypes between species may be attributed to several reasons, maybe if the authors use other molecular markers they were able to find genetic structure, besides, species are more than a genetic distance, I suggest to remove the taxonomic considerations of the *Harpagifer* species that are beyond the aim of the paper.

Lanes 353 to 358 " Despite their distribution in different regions, different *Harpagifer* species still share similar dietary and ecological constraints...." Where is the evidence of stomachal contents or ENM that support these statement? Recent divergence does not necessarily imply similar diets or PNC. Please change the redaction.

Minor comments

Lane 119 says geogianus must be georgianus

Why use Bray Curtis distance instead Unifrac which is based on a phylogeny? Please justify

Please use the name of the genera in the results instead of OTU number

Reviewer #2 (Comments for the Author):

This manuscript deals with the detection of phylo-symbiosis and co-phylogeny between *Harpagifer* species and their gut mucosal community in the Southern Ocean. It is well written, the material and methods is nicely detailed, the statistical analyses were carefully chosen and I appreciated the microdiversity approach. Yet, I have several concerns regarding the manuscript.

MAIN COMMENTS:

- 111 references for a manuscript of this length is excessive. Please limit this number to < 70-80. One possibility would be to shorten the Discussion, which is a bit long and contains many references.

- The environmental matrix does not seem very reliable to me: data were extracted from the BIO-ORACLE database as mean data for the period 2000-2014 (if I am correct). To which extent are these data representative of the physico-chemical environment at the time of sampling? Is this environment seasonally/yearly/... stable in a region undergoing rapid changes due to climate change? Why to have chosen "mean data at mean depth": at what depth do these 4 fish species live? These limitations are not discussed in the manuscript, whereas they should (at least briefly).
- Disclosing the existence of a core gut mucosal community (line 385): how did you choose a prevalence of 40% (which sounds pretty low)? If you had chosen another threshold, you might have concluded there was no core community at all... Plus, this core community was characterized by "low diversity" (line 36, line 387): "low" compared to what? The total microbial community of these Harpagifer species? The core gut mucosal microbiota of other fish species?
- Line 181: Before running a PERMANOVA, it is necessary to run a PERMDISP test to check whether groups differ in their variances. If the latter are different, a significant result of the PERMANOVA test (differences between groups) could be misinterpreted as the result of the factor tested, whereas it could be due to heterogeneity of multivariate variances. I do not find this PERMDISP test in the manuscript.
- Results part 3.2: The PERMANOVA test to check for the effect of fish species is weak but significant. If you had run this test on the location as variable, it would have likely been significant as well, as each of the four species is found solely in one location (the 4 WAP locations merged). So how to separate the host species and location effects?
- Line 40 and line 256: you mention a "robust" phyllosymbiosis signal. I do not agree at all. First, the (partial) Mantel correlations tests show that Environment, host phylogeny and geography all explain part of the variability in GMM of Harpagifer species (the highest contributor being environment, not host phylogeny). Second, robust testing of phyllosymbiosis usually involves different species found together in different locations. Here, each species is unique to one location (and indeed each fish species or corresponding location WAP/SOG/KER/PAT are used interchangeably throughout the manuscript). So again, how can you disentangle the effect of location (beyond the mean physico-chemical parameters presented - for example seawater or diet microbial community) and of host? This is my major concern regarding this manuscript. A more careful drafting of the abstract and discussion on this specific point is necessary.
- OTU2 and OTU4 are major core OTUs which are under-represented in the dataset compared to the predictions of the neutral model. One possibility is that "they are constrained by dispersal limitation" (line 292). What do you mean? Please explain (this point is not mentioned in the Discussion). Another explanation is the selection against by the host. In this context, could you provide some information from the literature about the relative abundance of Aliivibrio in Antarctica and Patagonian waters? It seems to be common in seawater in general (line 440).
- Table 3 and Figure 5C : there were fewer links between Harpagifer haplotypes from KER and OTU2 oligotypes. Could it be that the limited sampling in KER captured correctly the fish haplotype diversity but only partially the OTU2 Aliivibrio oligotype diversity (which is much higher)? This would bias the results.
- What is the diet of the 4 species? Besides the fact that they have "the same trophic positioning" (line 126), and a "similar diet" (line 356), the reader gets no information in this manuscript about their diet that could explain the dominance of (different) Aliivibrio oligotypes in the gut mucosal communities. Please elaborate; it would be useful to mention the proportion of crustaceans in the diet to support your hypothesis about the functional role of Aliivibrio (line 405-414).

MINOR COMMENTS :

- Number of fish individuals analyzed: 77 (line 30), 81 (line 239) or 78 (Table 1)? Please double check.
- Line 37: replace taxon (that could be defined at any taxonomic level) by OTU (i.e. 97% similarity in this study).
- Line 56: which increasing threats are these fish species facing in the Southern Ocean? Please explain. In addition, how would this study "help predict the consequences of environmental disturbances on the microbiome and host fitness"? it is unclear to me.
- Lines 69-72 ("yet directly testing..."): please rephrase, the message is unclear.
- Line 83: the sentence in brackets is useless and confusing (not all microbial lineages conforming the microbiome are "specific symbionts").
- Line 336: It is trivial that skin bacterial communities are more influenced by the surrounding environment than gut mucosal ones. Therefore I would restrain the discussion about the presence or absence of phyllosymbiosis to fish GUT communities.
- Figure 1: define APF and ACC.
- Figure 3: please explain the abbreviations on the X axis.
- Figure 5: footnote (4) is missing, the position of (5) in the Figure is odd
- Table 2: GMB = GMM? For the partial Mantel tests, the two factors mentioned are those controlled and not the one tested, if understood correctly. This presentation is misleading.
- Supplementary Figure 3: whenever possible (like in this figure), mention the exact p-values rather than " $p < 0.001$ ". Plus, explain better the specialist/generalist symbiont model in a few lines in the manuscript.
- Did you rarefy the 16S data, as you did in reference (53)? If so, to what number of sequences?
- Did you deposit your 16S data in the Short Read Archive?

SPELLING MISTAKES:

Line 65: metabolic, not metabolism
Line 104: microbiome, not microbiomes
Line 113: low or poor, not both
Line 183: why "respectively"?
Line 226: abundant, not abundance

We express our sincere gratitude to the reviewers for their thorough examination of our manuscript. We deeply appreciate their meticulous efforts.

In the following sections, we provide responses and explanations for each reviewer's comments.

Reviewers' comments:

Reviewer #1:

R1.Q1. In lanes 121 to 123 the authors state that "*The diversification of the Harpagifer genus occurred 1.2-0.8 Myr ago, from Antarctica towards the Patagonia and sub-Antarctic areas during the Pleistocene* ". The family Harpagiferidae comprise one genus and 12 species included in the genus *Harpagifer*. According to Matschiner the diversification of the more derived notothenioid families including Harpagiferidae begin in the mid Miocene (HPD 9.9-20 Ma). The authors suggest a recent divergence of their studied species *H. antarcticus* and *H. bispinis* based on the calibration of Hüne et al. who only include these two species. The authors include in their analysis other two species, *H. georgianus* that cannot be discriminated from *H. antarcticus* and *H. kerguelensis*. In the absence of a phylogeny for the complete 12 species of the genus, it is impossible to state if *H. kerguelensis* is, in reality, the sister clade of the *H. antarcticus* + *H. bispinis* clade, I agree with that, but the authors cannot suggest a recent divergence among the four studied species unless they build a phylogenetic calibrated tree that includes also *H. kerguelensis*.

We acknowledge that we extrapolated the conclusion of Hüne et al. regarding the recent divergence of H. bispinis and H. antarcticus to the two additional species in our study. The sequences of the other 8 species are not currently published/available, they are difficult to obtain (remote locations), and thus building the whole phylogeny will represent the objective of future studies.

In the context of the current study, in order to strengthen our claim concerning the recent divergence of Harpagifer, we calculated mean p-distance among the four nominal Harpagifer species used in this study detailed in Material and Methods section n°2.3, and we incorporated new Supplementary Tables 1 & 2. Results show low p-distances among species, ranging from 0.29% between H. antarcticus and H. georgianus to 0.96% between H. bispinis and H. kerguelensis. These values are particularly low when comparing to the classical study of Ward et al. (2009) (1), which reported Means and ranges of Kimura 2-parameter distance values by taxonomic rank among 1088 species of fish (0.34 ± 0.005 for intraspecific comparisons and 8.39 ± 0.023 for congeneric ones). Additionally, the haplotype network reveals that H. antarcticus shares three haplotypes with H. georgianus. Notably, a lone mutation distinguishes the H. kerguelensis haplogroup from the antarcticus-georgianus one, while a difference of three mutations distinguishes the H. bispinis haplogroup from the antarcticus-georgianus one. All these results strongly suggest a recent diversification of at least these four studied species. These results have been incorporated in the revised version of the manuscript, lines 280-295. We also tempered our statements about the recent diversification of Harpagifer in the abstract section, lines 27-29 and lines 42-44.

New Supplementary Table 1: Summary of haplotypes number and genetic indices per Harpagifer host species.

Host species	N	k	S	H	Π	π
H. antarcticus	42	7	7	0.7305 ± 0.0452	2.0340 ± 1.1660	0.003065 ± 0.001951
H. georgianus	7	5	5	0.9048 ± 0.1033	1.9047 ± 1.2281	0.002869 ± 0.002119
H. kerguelensis	11	6	5	0.8000 ± 0.1138	1.5636 ± 1.0076	0.002355 ± 0.001712
H. bispinis	21	10	12	0.7333 ± 0.1042	1.3905 ± 0.8893	0.002094 ± 0.001495

Number of sequences (N), number of oligotypes (k), number of polymorphic sites (S), genetic diversity (H), pairwise differences between sequences (Π) and nucleotide diversity (π) are detailed.

Supplementary Table 3: Summary of genetic p -distances (%) within and among the four *Harpagifer* species.

	H. bispinis	H. kerguelensis	H. georgianus	H. antarcticus
H. bispinis	0.2101	< 0.0001	< 0.0001	< 0.0001
H. kerguelensis	0.9636	0.2361	< 0.0001	< 0.0001
H. georgianus	0.5896	0.4541	0.2880	0.4636
H. antarcticus	0.9000	0.5173	0.2944	0.3080

The genetic p -distances within and among the four species are presented in, and below the diagonal, respectively. The p -values, determined through 10,000 permutations, are shown on the upper diagonal.

R1.Q2a. The authors report the collection of 77 individuals in Table 1, but in lane 239 they report that a total of 81 COI sequences were obtained, thus, the number of individuals and sequences does not match.

Indeed, there was a misunderstanding. We addressed the issue in lines 297-298: "Out of the 81 Harpagifer individuals sampled, 77 gut mucosa samples were successfully processed through metabarcoding, [...]". Please also see the answer to minor comment #1 of reviewer 2 for details.

R1.Q2b. On the other hand, they mention that a total of 25 haplotypes were recovered and formed three haplogroups of supplementary figure 2, but, the number of haplotypes in their figure 2 only comprise 22 haplotypes.

We double checked the number of haplotypes in supplementary figure 2, and we did count 25 haplotypes (indeed, haplotypes are named from Hap_1 to Hap_25).

R1.Q3. Regarding the haplotype network, the authors suggest the recognition of three haplogroups, PAT, KER, and WAP+SOG. At first sight, their appreciation appears correct, however, Hap 13 (PT) is separated from Hap 5(WAP+SOG) by three mutation steps, but Hap5 (WAP+SOG) is separated by only one mutation step from Hap22(KER), on the other hand, Hap12 (PAT) is separated by four mutations from Hap8(PAT). Due to the low divergence of the COI sequences and the apparent maintenance of ancestral polymorphism, is clear that a phylogenetic approach is not the best choice to support the number of haplogroups, but neither is an arbitrary separation based on a visual approach to the haplotype network. I recommend the authors the use of an approach to defining genetic clusters at the population level such as baps algorithm implemented in fastbaps library.

The primary objective of the haplotype network was to illustrate the relatively low genetic divergence of the four studied species, thus supporting their recent diversification (see previous answer to comment R1.Q1). However, we added statistical tests based on pairwise p -distance performing permutations of individuals among species to detect significant phylogeographic structure within our sampling.

*Our results showed highly significant differentiation among species, except for the *H. antarcticus* and *H. georgianus* comparison ($p = 0.46$), supporting the existence of three genetic groups (see materials and methods section 2.3 and result section 3.1). These results are now detailed in the section 3.1.*

R1.Q4a. The authors report an ML phylogenetic tree obtained with PhyML, they need to report which substitution model was used and how it was chosen, and whether they performed branch support? Please explain.

The required information was added to the methods section n°2.3 at lines 190-199.

R1.Q4b. Additionally, their phylogenetic tree presents many polytomies in opposition to the phylogenetic tree of *Allivibrio* oligotypes. The V3 and V4 primers of Klindworth allow to amplify an amplicon of 464 bp, if the authors used a 97% similarity threshold to discriminate among OTUs it means that all the oligotypes of *Allivibrio* must be concentrated in a range of a maximum of 14 bp, it surprises me how a total of 51 oligotypes could be recognized within the genus *Allivibrio* and lead to an almost dichotomic tree in opposition to the phylogenetic tree of *Harpagifer* using the same methodological approach (ML) that as mentioned earlier need to be explained at the detail (substitution model used and how it was chosen).

*We meticulously reexamined the dataset of OTU2 oligotypes associated with *Allivibrio*. The mean number of base differences among OTU2 oligotype sequences was found to be 4.20 ± 0.01 , with a mean p -distance of $0.01 \pm 4.36e-5$. To put this into context, considering that a 3% difference within a 464 bp fragment corresponds to 14 potentially different bases, it would theoretically result in $14e4 = 38,416$ different oligotypes. As such, we are currently unclear about the comment raised by reviewer #1.*

We have incorporated detailed information about the methodological approach for the tree reconstruction of bacterial oligotypes into the methods section (Section 2.6) at lines 250-251.

R1.Q5. The author aims to test the hypothesis of a co-phylogenetic signal between *Harpagifer* species and shared members of their gut microbiomes (i.e. core microbial taxa) (lanes 137 and 138), earlier in the introduction the authors state that cophylogenetic signal is validated by congruent topologies of host species and symbiotic phylogenies, I completely agree with their statement, however, this strong postulate is at the same time the major flaw of their paper. A phylogenetic tree allows to recover of genealogical relationships between ancestors and descendants, but, populations evolve in a different way as species. For example, the authors search to state cophylogenetic signal between haplotypes and microbiome, in this way, they have three haplotypes shared between two species, *H. antarcticus*, and *H. georgianus*, the fact that both species shared the same haplotype could be the result of maintenance of ancestral polymorphism or incomplete lineage sorting among others factors. Despite sharing the same COI sequence both species have different evolutionary histories, if the authors want to track the coevolutionary trajectories of host and associates they will need the complete monophyletic group of 12 species of the genus *Harpagifer*, I know that this escape from the main goal and I do not suggest to make it, but I suggest that the authors need to perform the cophylogenetic tests for each one of the four species and not for the complete dataset, at the same time I suggest the authors to "ease up" some of the sentences of their introduction

due to their impossibility to include all the species included in the genus Harpagifer or at least they warn of the situation and limitation of their findings and conclusions.

*By definition, the cophylogeny is the study of “the concordance between the phylogenies and the interaction of two groups of associated species” (3), which differs from the phyllosymbiosis that refers to a congruence pattern between the phylogeny of host species and the clustering of microbial community structure (2). Consequently, these patterns do require different host species (e.g. haplotypes conforming different species, (4)) to be tested. Moreover, each one of our species is endemic to its respective biogeographic province (i.e. no overlapping distribution). Since the COI sequence was not resolute enough to discriminate *H. antarcticus*, and *H. georgianus*, we did not interpret the result data separately for these two species. In this context, we are uncertain about the specific request of reviewer #1 to “perform the cophylogenetic tests for each one of the four species”.*

While we fully concur with reviewer #1 observation regarding the limited number of species in our study, we would like to underscore the effort and the scientific value associated to the sampling. Indeed, it covers one-third of the species conforming the host genus, and had involved extensive national/international collaborations along with multiple scientific expedition to some of the most logistically challenging locations on Earth.

*Nevertheless, to address this limitation explicitly, we have incorporated a sentence in the conclusion of the manuscript, lines 530-540, along with reviewer #1 comment about the possible underlying reasons of *H. antarcticus*, and *H. georgianus* haplotypes sharing. We also eased up the hypothesis stated in the abstract, lines 27-29, and some sentences in the introduction dealing with Harpagifer diversification, lines 128-133.*

R1.Q6. The Mantel test was performed among GMM against host phylogeny, geography, and environmental distance matrices. The authors report a phylogenetic tree based on haplotypes but I suspect based on the results of the supplementary material 2 that the distance matrix for Mantel comparison of host phylogeny was performed only for the four species of the genus Harpagifer. It is not clear how this analysis was performed, if the authors used the four terminal species they need to explain how they estimated the GMM for each one of the Harpagifer species, mainly that they used Bray Curtis distance that is based on abundances, I assume that they had a GMM for each individual, for example, in *H. antarcticus* they have 38 individuals, how they obtain an average GMM for this species? On the other hand, if *H. antarcticus* and *H. georgianus* share the same haplotypes the genetic distance between this species must be zero, unless they use as genetic distance the private haplotypes for each species (HAP 24) for *H. georgianus* and HAP4, 6 or 7 for *H. antarcticus*, I am not able to understand how the authors perform this analysis, please explain at detail.

We agree with reviewer #1 that this part of the methodology section was unclear in the first version of our manuscript. Each sample corresponds to one Harpagifer individual. For each of the 77 samples, we have gut microbial community composition data (thousands of OTUs) and one host sequence (COI) data.

*For testing the phyllosymbiosis signal, the input of the Mantel test is a microbial community distance matrix (77*77 samples) obtained from the microbial OTU table (77 samples * 9782 OTUs) by Bray Curtis calculation: so, the GMM is not averaged*

by host species. Correlations are investigated between the GMM distance matrix and each one of the three explanatory matrices (spatial distance, environmental distance, host phylogenetic distance). More specifically, the host phylogenetic distances were obtained from the reconstructed phylogenetic tree of the 77 individual COI sequences.

For testing the co-phylogenetic signal, the input data of the Parafit and PACo tests are the phylogenetic tree of the OTU2 microdiversity (109 oligotypes) and the host phylogenetic tree (all the 25 haplotypes, and not the private haplotypes only).

*In the case of *H. antarcticus* and *H. georgianus* mentioned by the reviewer #1, all the haplotypes are not shared, thus their genetic distances are not null and contribute to the co-phylogenetic signal evidenced. As the COI gene is not resolutive enough to clearly discriminate *H. antarcticus* and *H. georgianus*, we graphically highlighted these two species with the same color (blue clade) in the Figure 5C. We have revised section 2.3 of the methods to provide a clearer and more concise explanation, at lines 190-199.*

R1.Q7. In lanes 332 to 338 the authors state "PACo analysis of Aliivibrio oligotypes (OTU2) revealed that the 'r2' model led to the highest phylogenetic congruence between host and microbe phylogenies, suggesting that the Aliivibrio phylogeny is driven by Harpagifer phylogeny (and not the opposite), and the degree of specialization of Aliivibrio oligotypes (quantified by the number of associations with Harpagifer haplotypes) also contributed to the global fit of the cophylogeny model " I agree with the fact that there is a clade of Aliivibrio oligotypes that extend from O248 to O717 that are exclusive of Harpagifer bispinis, nevertheless, these set of oligotypes form a polytomy at the base of the cladogram with the rest of the clusters present in the other Harpagifer species, so, the first split between Allivibrio oligotypes cannot be linked to the split between Harpagifer species, please change the redaction.

*It seems that the cophylo function in the phytools R package encountered difficulty plotting small phylogenetic branches, resulting in the appearance of polytomies. To address this graphical issue in the tanglegram, we transformed both host and symbiont trees into ultrametric trees after testing for co-phylogeny. Figure 5C now distinctly illustrates the presence of two Aliivibrio clades mirroring the *H. bispinis* / *H. antarcticus* ones. To improve graphical resolution, the 3 panels of the Figure 5 have been reorganized.*

On the other hand, we acknowledge that the split between KER and WAP/SOG host species are not followed by the symbionts, probably due to an insufficient resolution associated to the slow mutation rate of the 16S rRNA gene. We briefly discussed this limitation in the revised version of the manuscript, lines 530-534.

R1.Q8a. In the discussion lanes 361 to 363 the authors state "the gut microbiomes were more similar among individuals of the same Harpagifer species than among different species", however, in their results, they state in lanes 249 to 252 that "GMM comparisons were different except between *H. antarcticus* and *H. georgianus* mirroring the absence of genetic differentiation among these species", [...]

*We agree that these statements may sound contradictory and we rephrased it accordingly, lines 425-429: "In other terms, when conducting pairwise comparisons among Harpagifer species—except for the comparison between *H. georgianus* and*

H. antarcticus—the gut microbiomes show greater similarity among individuals of the same species compared to individuals of different species, and these similarities are congruent with the branching pattern of the species phylogeny.”

R1.Q8b. [...] meanwhile, in lanes 253 and 254 they suggest that the weak differences in PERMANOVA reflect the similar seawater properties of both regions. I am confused, in the abstract the authors state that the core microbiome of *Harpagifer* is characterized by a low diversity mainly driven by selective processes, I assume that if a phylosymbiosis signal is present the selective pressures are imposed by the host as suggested in lanes 389 to 393 and not by the environmental water parameters. I suggest the authors carefully review these sentences that appear contradictory.

In lines 253-254, we present findings related to the total gut microbiome, whereas in L 389-393 we shifted specifically to the shared OTUs among the four Harpagifer species microbiomes (i.e. the core). Importantly, we want to clarify that we did not claim the phylosymbiosis signal driven by the host as the exclusive factor shaping the Harpagifer gut microbiome. Indeed, the surrounding environment properties also partly contribute to the dissimilarity in microbiome composition (as supported by the partial mantel tests). However, since it is composed of OTUs common to the different host species, we can expect that the core microbiome contains bacterial OTUs that may be more influenced by host-related factors. Therefore, we believe these statements are not contradictory but rather complementary, providing a nuanced understanding of the various factors influencing the Harpagifer gut microbiome.

R1.Q9. Why to use the r2 option in Paco? Procrustes analysis of cophylogeny allows performing the analysis through different randomization approaches including r2 model. In their discussion, the authors state that " the most likely co-phylogenetic model was the adaptive tracking of *Harpagifer* phylogeny by *Allivibrio*..." Did the authors perform the PACo analysis using all randomization options? How do they compare them and choose the r2 as the best predictor? Their selection of r2 model must be fully justified in the methods.

The selection of the randomization method in PACo was described in the original version of the manuscript in lines 228-233, and results were presented in Supplementary Figure 3. Specifically, we tested 4 randomization models, including r0 referred as ‘symbiont model’ (assuming the symbionts group tracks the hosts’ evolution), c0 referred as ‘host model’ (assuming the hosts’ group tracks the symbionts evolution), ‘quasiswap’ referred as ‘undetermined’ (unclear which group tracks the other) and r2 referred as ‘specialist/generalist symbiont’ (assuming the symbionts’ group tracks the hosts’ evolution and the specialization/generalism of symbionts, i.e. number of partners, also determines cophylogenetic signal). The best randomization model was determined by comparing the values of phylogenetic congruence for each model obtained through 20,000 iterations (Supplementary Figure 3).

To address reviewer #1 concern, we added more information in the methods section at lines 263-270: “Four randomization algorithms were tested in PACo— ‘r0’, ‘c0’, ‘quasiswap’ and ‘r’ models. The ‘r0’ model (referred as ‘symbiont’) posits that symbionts track hosts’ evolution, while the ‘c0’ model (referred as ‘host’) assumes that hosts track symbionts’ evolution. The ‘quasiswap’ model (referred as ‘undetermined’) does not infer direction in the tracking, and the ‘r2’ model (referred as ‘specialist/generalist symbiont’) posits that symbionts track hosts’ evolution, with the

co-phylogenetic signal influenced by the specialist/generalist feature of the symbionts (i.e. number of partners)."

R1.Q10. In lanes 466 to 469 the authors state" the absence of genetic and phylogeographic structure between SOG and WAP question the validity of *H. antarcticus* and *H. georgianus*". I am not a specialist in the group and the information provided does not allow me to track the taxonomy of both species, but, Eschemeyer fish catalog recognizes both species as valid. Both, population genetics and phylogeography have their epistemology, but the analysis used by the authors does not correspond to either, moreover, as I mentioned several times, shared haplotypes between species may be attributed to several reasons, maybe if the authors use other molecular markers they were able to find genetic structure, besides, species are more than a genetic distance, I suggest to remove the taxonomic considerations of the Harpagifer species that are beyond the aim of the paper.

We agree with reviewer #1 that this paragraph goes beyond the aims of the paper, and we removed it accordingly.

R1.Q11. Lanes 353 to 358 " Despite their distribution in different regions, different Harpagifer species still share similar dietary and ecological constraints...." Where is the evidence of stomachal contents or ENM that support these statements? Recent divergence does not necessarily imply similar diets or PNC. Please change the redaction.

We rephrased this discussion sentence to center on the diet similarity, as several studies have previously described the stomach contents of several of the Harpagifer species involved in the present work, at lines 417-421: "This result suggests that, despite their distribution in distinct biogeographic regions, different Harpagifer species still share similar ecology leading to relatively similar gut microbial communities, as expected in recent allopatric speciation scenario (6, 16), and in lines with existing knowledge regarding the dietary consistency among these species (33, 34)."

Minor comments

- Lane 119 says geogianus must be georgianus.

We modified accordingly.

- Why use Bray Curtis distance instead Unifrac which is based on a phylogeny? Please justify.

To demonstrate microbiome distinguishability among species and to assess the correlation between host phylogeny and microbiome compositions, several algorithms for distance matrix calculation are classically used, including Jaccard, Bray-Curtis, or Unifrac, all being considered valid (2). We opted for the most commonly used metric in microbiome studies to enhance comparability with existing literature. Additionally, we selected a metric that is computationally efficient for implementation in our R code. Specifically, using Unifrac would necessitate the computation of a phylogenetic tree for all bacterial taxa within the microbiome, imposing a substantial computational burden.

Notwithstanding these considerations, we used the weighted Unifrac distances in the mantel tests; we found that each of the three explanatory matrices accounts for the same amount of microbiome composition variance ($0.18 < R^2 < 0.19$), with the host

phylogeny having the slightly highest R^2 value (cf table below). These results were added to the revised version of the manuscript, under a new supplementary table n°4, and cited in the result section 3.3, lines 321-324: “When dissimilarities of Harpagifer GMM were calculated using the weighted UniFrac metric, the degrees of correlation were lower and homogeneous across the explanatory matrices, remaining statistically significant (Supplementary Table 5).”

New Supplementary Table n°5. Mantel test analysis on GMM of Harpagifer spp. using weighted UniFrac distances.

Factors	Statistical test	R²	p-value
Environmental distance	Mantel	0.18	<0.001
Geographic distance	Mantel	0.18	<0.001
Host phylogeny	Mantel	0.19	<0.001
Environmental Geographic	Partial mantel	0.11	<0.001
Environmental Host phylogeny	Partial mantel	0.05	<0.05
Geographic Environmental	Partial mantel	0.12	<0.001
Geographic Host phylogeny	Partial mantel	0.13	<0.001
Host phylogeny Environmental	Partial mantel	0.14	<0.001
Host phylogeny Geographic	Partial mantel	0.05	<0.001

We also performed the PERMANOVA with weighted UniFrac metric to test the global effect of host species identity on Harpagifer microbiome composition. The results, identical to those obtained with Bray-Curtis were added to the result section 3.3, lines 299-302: “A weak but significant effect of Harpagifer species identity on GMM composition was detected through PERMANOVA using both Bray-Curtis (F -statistics=4.46, $R^2=0.15$, $p<0.001$) and weighted UniFrac metrics (F -statistics=4.33, $R^2=0.15$, $p<0.001$).”

- Please use the name of the genera in the results instead of OTU number.

Considering that OTU2 accounted for more than 99% of the sequences affiliated with Aliivibrio in the GMM, we referred to this OTU using the genus name and we added the corresponding justification lines 352-354: “A total of 393,084 sequences, constituting 51% of the GMM, were affiliated with the Aliivibrio genus. Notably, OTU2 accounted for 99.4% of these sequences, and consequently, is referred to hereafter as Aliivibrio.”

Reviewer #2:

Main comments:

R2.Q1. 111 references for a manuscript of this length is excessive. Please limit this number to < 70-80. One possibility would be to shorten the Discussion, which is a bit long and contains many references.

To address the concerns raised by reviewer #2, we pruned a total of 23 references from the initial version of the manuscript, resulting in an approximate 21% reduction, leaving 88 references in the revised version.

Considering:

- (i) *The multidisciplinary nature of our study (i.e. many references are associated to specific the methodologies of macro- and microorganisms).*
- (ii) *The absence of specific reference number restrictions in the guidelines of Microbiology Spectrum (some article even exceeds 150 references; <https://doi.org/10.1128/spectrum.02436-23>).*
- (iii) *The numerous additional inputs sought by both reviewers (requesting the incorporation of new references).*

We kindly request the reviewer 2 to reconsider the suggested threshold of 70-80 references. In the meantime, we removed a whole paragraph of the discussion section (> 100 words) about the taxonomic considerations of Harpagifer species (see answer to comment R1.Q10).

R2.Q2. *The environmental matrix does not seem very reliable to me: data were extracted from the BIO-ORACLE database as mean data for the period 2000-2014 (if I am correct). To which extent are these data representative of the physico-chemical environment at the time of sampling?*

Is this environment seasonally/yearly/... stable in a region undergoing rapid changes due to climate change? Why to have chosen "mean data at mean depth": at what depth do these 4 fish species live? These limitations are not discussed in the manuscript, whereas they should (at least briefly).

Due to substantial geographical separation among sites, and the challenging access to these remote sites, sampling at the same time was unfeasible (see Table 1). To address this constraint, we decided to use the BIO-ORACLE database, which spans all our sampling sites and provides averaged data over an extended period (i.e., 2000-2014), allowing us to account for seasonal and annual variability. This database, widely acknowledged and extensively applied in studies involving both macroorganisms and microorganisms across the Southern Ocean and other oceanic regions, enjoys broad acceptance within the scientific community (5-8).

An alternative approach would have been to measure physico-chemical parameters at the time of sampling. However, it would not have reliably capture the temporal variability of these parameters. Hence, we chose a more integrative approach.

Concerning the depth, Harpagifer species are benthic and inhabit shallow waters, spanning within a range 0–20 m (9). Due to the logistical challenge of sampling all the individuals from a same site at the same depth, we specifically relied on mean depth values to mitigate inherent variability in individual sampling depths.

However, we agree with reviewer #2 that these methodological limitations should be clearly mentioned in the manuscript, and are now addressed in the Methods section, lines 210-215.

R2.Q3. *Disclosing the existence of a core gut mucosal community (line 385): how did you choose a prevalence of 40% (which sounds pretty low)? If you had chosen another threshold, you might have concluded there was no core community at all...*

The purpose of detecting a core microbiome common to all Harpagifer species is to pre-filter the taxa that are potentially relevant for further analysis (i.e. here, to detect a co-phylogeny signal). Therefore, the choice of a core threshold is always suggestive,

at the discretion of the authors, and the cut off usually ranges between 30-100% of prevalence across samples (10, 20).

The choice of core thresholds has been extensively reviewed in the literature. For instance, A. T. Neu et al. (20): "The proportions of sites, samples, or time points over which a microbe must occur to be considered core, however, is always at the discretion of the authors. In the most liberal cases from our dataset, a 30% occurrence standard was used, meaning that any OTU detected in at least 30% of samples was considered a core member (17, 28). Others have used cut offs between 50 and 99.9%, depending on the study system and number of total samples".

Our study falls within that range, by using 40% prevalence cut off across all gut mucosa samples of the dataset. More details about the number of core OTUs according to the prevalence threshold can be deduced from the column "Prev." in the Table 3 (e.g. at 80% prevalence, 7 core OTUs were detected).

Plus, this core community was characterized by "low diversity" (line 36, line 387): "low" compared to what? The total microbial community of these *Harpagifer* species? The core gut mucosal microbiota of other fish species?

We intended to convey that the observed low diversity was in comparison to the total gut mucosal microbiota of *Harpagifer*. Specifically, the 17 core OTU represent less than 0.2% of the total number of OTU of the gut mucosal microbiota. We added this precision to the statements at lines 334-337 "Constituting less than 0.2% of the total OTU richness in GMM, this core microbiome exhibited a relative abundance of $22.5\% \pm 2.9\%$ in gut mucosa samples, encompassing representatives from 9 bacterial classes, predominantly within the Gammaproteobacteria phylum (Table 3)." and at lines 447-449 "This core was characterized by a relatively low diversity compared to the total OTU richness in GMM, and by the high dominance of a single taxon, as previously reported in the core microbiomes of other fishes (68-70)."

R2.Q4. Line 181: Before running a PERMANOVA, it is necessary to run a PERMDISP test to check whether groups differ in their variances. If the latter are different, a significant result of the PERMANOVA test (differences between groups) could be misinterpreted as the result of the factor tested, whereas it could be due to heterogeneity of multivariate variances. I do not find this PERMDISP test in the manuscript.

As a non-parametric test, PERMANOVA test is based on permutations, so it actually does not assumed homogeneity of variance, as clarified here:

<https://uw.pressbooks.pub/appliedmultivariatestatistics/chapter/permanova/>

"PERmutational Multivariate ANalysis of VAriance (PERMANOVA) is a permutation-based technique – it makes no distributional assumptions about multivariate normality or homogeneity of variances".

In order to address any concerns from Reviewer 2, we performed a PERMDISP test using the *betadisp* function implemented in the R package *Vegan*, which turns out to be not significant (F -statistic = 0.6896, p = 0.5614), indicating that groups do not differ in dispersion, so our conclusions from PERMANOVA are valid.

R2.Q5. Results part 3.2: The PERMANOVA test to check for the effect of fish species is weak but significant. If you had run this test on the location as variable, it would have likely been significant as well, as each of the four species is found solely in one location (the 4 WAP locations merged). So how to separate the host species and location effects?

Instead of demonstrating the influence of Harpagifer species on microbiome composition, our objective was to assess the distinguishability of microbial communities among the four species. In other words, we sought to test whether intraspecies variability was lower than interspecies variability, without solely attributing this variability to the host factor. This is a mandatory prerequisite to robustly demonstrate the existence of a phylosymbiosis signal (2, 10).

In the next Results section (3.3.) we precisely try to separate the host species and location effects, as asked by Reviewer 2, through partial Mantel tests. They show that the effect of host phylogeny (controlling for geography) is higher than the effect of geography (controlling for host phylogeny) (Table 2).

R2.Q6. Line 40 and line 256: you mention a "robust" phylosymbiosis signal. I do not agree at all. First, the (partial) Mantel correlations tests show that Environment, host phylogeny and geography all explain part of the variability in GMM of Harpagifer species (the highest contributor being environment, not host phylogeny). Second, robust testing of phylosymbiosis usually involves different species found together in different locations. Here, each species is unique to one location (and indeed each fish species or corresponding location WAP/SOG/KER/PAT are used interchangeably throughout the manuscript). So again, how can you disentangle the effect of location (beyond the mean physico-chemical parameters presented - for example seawater or diet microbial community) and of host? This is my major concern regarding this manuscript. A more careful drafting of the abstract and discussion on this specific point is necessary.

When referring to a "robust" signal, we do not mean "predominantly driven by the host", but we rather intend to convey its statistical significance, in contrast to the conventional significance of phylosymbiosis signals observed in wild fish microbiomes (11, 12). Nevertheless, it is worth nothing that when using weighted Unifrac method for microbiome composition distances, the host factor was slightly higher than environmental and geographic distances (see answer to reviewer #1 minor comment). To address reviewer #2 concern, we changed "robust" for "significant" in the title of Result section 3.3.

Moreover, we would like to emphasize that phylosymbiosis is "agnostic to mechanism" (13). This pattern could be the result of host evolutionary history, shared ecology derived from phylogenetically conserved host traits (i.e. diet, habitat, host immune systems), vertical transmission of microbiome, etc (14). Consequently, an overlapping distribution of different species found together in different locations is not mandatory for phylosymbiosis testing; see, for instance, references (15), (10), (16) and (17). These precisions were added to the first paragraph of the discussion section, lines 406-410. In summary, phylosymbiosis relies on two fundamental prerequisites: (i) the distinguishability of microbiome composition among host species and (ii) a correlation between microbiome composition and host phylogenetic distances. Both criteria were met with substantially low p-values, underscoring a robust phylosymbiosis signal.

Concerning the specific concern about the respective effect of environment and host, please see previous answer to comment R2.Q5.

R2.Q7a. OTU2 and OTU4 are major core OTUs which are under-represented in the dataset compared to the predictions of the neutral model. One possibility is that "they

are constrained by dispersal limitation" (line 292). What do you mean? Please explain (this point is not mentioned in the Discussion).

We rephrased this sentence of the result section, providing brief explanations, lines 347-352: "The most abundant and prevalent OTU in the core microbiome of Harpagifer, namely OTU2 affiliated with Aliivibrio, exhibited a lower frequency (and higher abundance) than the 95% confidence interval predicted by the neutral model, suggesting that this OTU is either selected against by the host (i.e., invasive pathogenic taxa) or experience significant dispersal limitations (i.e., low probability of successful host colonization)."

R2.Q7b.

Another explanation is the selection against by the host. In this context, could you provide some information from the literature about the relative abundance of *Aliivibrio* in Antarctica and Patagonian waters? It seems to be common in seawater in general (line 440).

To the best of our knowledge, there is currently no published information about the relative abundance of Aliivibrio in Southern Ocean seawater. However, this genus has been reported in various other Antarctic and sub-Antarctic animals such as sponges (18) and toothfishes (19). Moreover, as other Vibrionaceae, is predominantly found attached to zooplankton rather than existing in a free-living state in seawater (20), possibly explaining its facile colonization of Harpagifer gut. We acknowledge that our initial sentence may have been ambiguous and could have been misconstrued to suggest the ubiquity of Aliivibrio in seawater. We rephrased to clarify, lines 502-508: "Although specific data on the relative abundance of planktonic Aliivibrio in the Southern Ocean are lacking, a plausible explanation for the widespread detection of Aliivibrio sp. in fishes is the frequent colonization of new individuals within a same fish species through the excretion of Aliivibrio-rich faeces into the surroundings (74). Moreover, as other Vibrionaceae, Aliivibrio is predominantly found attached to zooplankton rather than existing in a free-living state in seawater (85), possibly explaining its facile colonization of Harpagifer gut."

R2.Q8. Table 3 and Figure 5C: there were fewer links between Harpagifer haplotypes from KER and OTU2 oligotypes. Could it be that the limited sampling in KER captured correctly the fish haplotype diversity but only partially the OTU2 *Aliivibrio* oligotype diversity (which is much higher)? This would bias the results.

To confirm the robust coverage of OTU2 Aliivibrio oligotypes across Harpagifer species, we generated the rarefaction curves of OTU2 microdiversity according to three alpha diversity metrics (Observed richness, Chao1 and ACE, cf. figure below). Across the three metrics, rarefaction curve profiles of oligotype sequences were globally consistent and almost reached saturation across Harpagifer species, confirming the good coverage of the microdiversity within OTU2 Aliivibrio and the validity of the cophylogenetic pattern detailed in Figure 5.

Rarefaction curves of OTU2 microdiversity across Harpagifer species according to three alpha diversity metrics.

R2.Q9. What is the diet of the 4 species? Besides the fact that they have "the same trophic positioning" (line 126), and a "similar diet" (line 356), the reader gets no information in this manuscript about their diet that could explain the dominance of (different) *Aliivibrio* oligotypes in the gut mucosal communities. Please elaborate; it would be useful to mention the proportion of crustaceans in the diet to support your hypothesis about the functional role of *Aliivibrio*.

The only relevant information about Harpagifer diet currently available in the literature is already provided in the discussion of the first version of the manuscript (lines 408-410) and deals with chitin-rich crustaceans accounting between 70% and 100% of the diet biomass of H. bispinis, H. antarcticus and H. georgianus. We rephrased the lines 472-476 of the discussion in the new version of the manuscript, to integrate the proportion of crustaceans: "These crustaceans are dominant in the diet of H. antarcticus, H. georgianus and H. bispinis, comprising between 70% and 100% of the biomass ingested (35, 36). Additionally, Aliivibrio, along with other together with other Vibrionaceae, are common members of crustacean microbiota, such as copepods (81)."

Minor comments:

- Number of fish individuals analyzed: 77 (line 30), 81 (line 239) or 78 (Table 1)? Please double check.

We revised Table 1 and it did indicate 77 (and not 78) gut mucosa. The total number of sampled Harpagifer individuals was 81 individuals. Although successful amplification of the CO1 gene was achieved for all 81 individuals (presented in host haplotype network), the gut mucosa amplification was successful for 77 individuals only. To address potential confusion for readers, we have made the necessary corrections by updating Table 1 to reflect the accurate count of 81 fish individuals. Additionally, we have modified a statement in the section 3.2 to explicitly clarify the distinction between the total number of individuals sampled (n = 81) and the subset

with successful gut mucosa amplification (77), lines 297-298: "Out of the 81 Harpagifer individuals sampled, 77 gut mucosa samples were successfully processed through metabarcoding, [...]".

- Line 37: replace taxon (that could be defined at any taxonomic level) by OTU (i.e. 97% similarity in this study).

We modified accordingly.

- Line 56: which increasing threats are these fish species facing in the Southern Ocean? Please explain. In addition, how would this study "help predict the consequences of environmental disturbances on the microbiome and host fitness"? it is unclear to me.

Marine fishes, particularly stenothermic species adapted to the cold waters of the Southern Ocean, such as the notothenioid species Harpagifer spp., are notably susceptible to the impacts of climate change (21-23). Indeed, in warmer waters, their specific adaptation to cold waters could represent a disadvantage. Specifically, several studies have demonstrated the sensitivity of Harpagifer bispinis and Harpagifer antarcticus to temperature increase, salinity decrease, and quantified their exposition to microplastic contamination (24-27). In this context, exploring potential evolutionary interdependence between the Harpagifer host and its specific symbionts could unveil microbial candidates worthy of future monitoring. Such symbionts may play a pivotal role, either contributing to or limiting the acclimatization of host species to a rapidly changing environment (28). To address reviewer #2 concern, we rephrased the whole "Importance" section, lines 46-59 and added two sentences to the Introduction section, lines 121-128: "These species are stenothermic, demonstrating an adaptation to cold waters, and have been identified as susceptible to the impacts of climate change (e.g. seawater temperature increase) and anthropogenic perturbation (i.e. microplastic contamination) in the Southern Ocean (31, 32). In this context, exploring potential evolutionary interdependence between Harpagifer and its specific symbionts could unveil microbial candidates for future monitoring. These symbionts may play a crucial role in either contributing to or limiting the acclimatization of host species to a rapidly changing environment (33)."

- Lines 69-72 ("yet directly testing..."): please rephrase, the message is unclear.

We rephrased the sentence, adding more details about the methodological challenges related to co-diversification testing, lines 69-73: "However, demonstrating co-diversification between hosts and complex microbiomes in natural populations remains methodologically challenging, because signal might be weak and/or overwhelmed by environmental factors, and insufficient genomic data hinder the determination of divergence timing and the acquisition of well-resolved phylogenies (8, 9)."

- Line 83: the sentence in brackets is useless and confusing (not all microbial lineages conforming the microbiome are "specific symbionts").

We modified accordingly.

- Line 336: It is trivial that skin bacterial communities are more influenced by the surrounding environment than gut mucosal ones. Therefore, I would restrain the discussion about the presence or absence of phyllosymbiosis to fish GUT communities.

Since line 336 did not address fish skin bacterial communities, we inferred that reviewer #1 was referencing line 366. We have appropriately removed the corresponding sentences and rephrased this part of the discussion, lines 430-433: "Studies investigating phyllosymbiosis in the gut microbiome of anciently diverged wild marine fish are sparse, and the reported cases vary from absent to moderate signal, with a limited understanding of the factors that influence its intensity (22, 62, 63)."

- Figure 1: define APF and ACC.

We precise the signification of APF and ACC in figure 1 legend.

- Figure 3: please explain the abbreviations on the X axis.

The signification of the abbreviations of the x-axis was added in the legend of the new version of the figure 3, see lines 863-865.

- Figure 5: footnote (4) is missing, the position of (5) in the Figure is odd

Footnote (4) was present in the first version of the Figure 5 (at the same position with footnote (2)). We updated the positions of footnote (4) and footnote (5) in a new version of the Figure 5.

- Table 2: GMB = GMM? For the partial Mantel tests, the two factors mentioned are those controlled and not the one tested, if i understood correctly. This presentation is misleading.

We modified accordingly the title of Table 2 and we improved the legend to address reviewer #2 concern (see lines 894-897).

- Supplementary Figure 3: whenever possible (like in this figure), mention the exact p-values rather than " $p < 0.001$ ". Plus, explain better the specialist/generalist symbiont model in a few lines in the manuscript.

All p-values in the comparison statistics linked to Supplementary Figure 3 were identical. To preserve figure clarity, we have included the p-value in the legend, denoting a significance level below $2.22e-16$.

The four randomization algorithms tested in PACo were explained in lines 263-270.

- Did you rarefy the 16S data, as you did in reference (53)? If so, to what number of sequences?

Yes, we did rarefy the 16S data as mentioned in section 2.4 of the original version of the manuscript: "The bacterial OTU table was rarefied at 5,750 sequences and converted into Bray-Curtis dissimilarity distances."

- Did you deposit your 16S data in the Short Read Archive?

Yes, we did deposit our 16S data, as mentioned in section 7 of the original version of the manuscript: "Amplicon sequences of 16S rRNA and COI have been deposited in the National Center for Biotechnology Information (NCBI), under the Sequence Read Archive (SRA) PRJNA803378, and in GenBank under the accession number ON891147 to ON891171, respectively."

Spelling mistakes:

Line 65: metabolic, not metabolism

We modified accordingly.

Line 104: microbiome, not microbiomes

We modified accordingly.

Line 113: low or poor, not both

We modified accordingly.

Line 183: why "respectively"?

We corrected the spelling mistake.

Line 226: abundant, not abundance

We modified accordingly.

References:

1. Ward RD. 2009. DNA barcode divergence among species and genera of birds and fishes. *Molecular ecology resources* 9:1077-1085.
2. Lim SJ, Bordenstein SR. 2020. An introduction to phylosymbiosis. *Proc Biol Sci* 287:20192900.
3. Blasco-Costa I, Hayward A, Poulin R, Balbuena JA. 2021. Next-generation cophylogeny: unravelling eco-evolutionary processes. *Trends Ecol Evol* 36:907-918.
4. Johnston EC, Cunning R, Burgess SC. 2022. Cophylogeny and specificity between cryptic coral species (*Pocillopora* spp.) at Mo'orea and their symbionts (Symbiodiniaceae). *Mol Ecol* 31:5368-5385.
5. Schwob G, Segovia NI, González-Wevar C, Cabrol L, Orlando J, Poulin E. 2021. Exploring the Microdiversity Within Marine Bacterial Taxa: Toward an Integrated Biogeography in the Southern Ocean. *Frontiers in Microbiology* 12:703792.
6. Frugone MJ, López ME, Segovia NI, Cole TL, Lowther A, Pistorius P, Dantas GP, Petry MV, Bonadonna F, Trathan P. 2019. More than the eye can see: Genomic insights into the drivers of genetic differentiation in Royal/Macaroni penguins across the Southern Ocean. *Molecular phylogenetics and evolution* 139:106563.
7. Aubert A, Beauchard O, de Blok R, Artigas LF, Sabbe K, Vyverman W, Martínez LA, Deneudt K, Louchart A, Mortelmans J, Rijkeboer M, Debusschere E. 2022. From Bacteria to Zooplankton: An Integrative Approach Revealing Regional Spatial Patterns During the Spring Phytoplankton Bloom in the Southern Bight of the North Sea. *Frontiers in Marine Science* 9.
8. Segovia NI, Gonzalez-Wevar CA, Haye PA. 2020. Signatures of local adaptation in the spatial genetic structure of the ascidian *Pyura chilensis* along the southeast Pacific coast. *Sci Rep* 10:14098.
9. Eastman JT. 1991. Evolution and diversification of Antarctic notothenioid fishes. *American Zoologist* 31:93-110.
10. Mazel F, Davis KM, Loudon A, Kwong WK, Groussin M, Parfrey LW. 2018. Is host filtering the main driver of phylosymbiosis across the tree of life? *Msystems* 3:e00097-18.
11. Sadeghi J, Chaganti SR, Johnson TB, Heath DD. 2023. Host species and habitat shape fish-associated bacterial communities: phylosymbiosis between fish and their microbiome. *Microbiome* 11:258.

12. Chiarello M, Auguet JC, Bettarel Y, Bouvier C, Claverie T, Graham NAJ, Rieuvilleneuve F, Sucre E, Bouvier T, Villeger S. 2018. Skin microbiome of coral reef fish is highly variable and driven by host phylogeny and diet. *Microbiome* 6:147.
13. Grond K, Bell KC, Demboski JR, Santos M, Sullivan JM, Hird SM. 2020. No evidence for phyllosymbiosis in western chipmunk species. *FEMS Microbiol Ecol* 96.
14. Qin M, Jiang L, Qiao G, Chen J. 2023. Phyllosymbiosis: The Eco-Evolutionary Pattern of Insect-Symbiont Interactions. *Int J Mol Sci* 24.
15. Groussin M, Mazel F, Sanders JG, Smillie CS, Lavergne S, Thuiller W, Alm EJ. 2017. Unraveling the processes shaping mammalian gut microbiomes over evolutionary time. *Nat Commun* 8:14319.
16. Tang Y, Ma KY, Cheung MK, Yang CH, Wang Y, Hu X, Kwan HS, Chu KH. 2021. Gut Microbiota in Decapod Shrimps: Evidence of Phyllosymbiosis. *Microb Ecol* 82:994-1007.
17. Dietrich C, Kohler T, Brune A. 2014. The cockroach origin of the termite gut microbiota: patterns in bacterial community structure reflect major evolutionary events. *Appl Environ Microbiol* 80:2261-9.
18. Savoca S, Lo Giudice A, Papale M, Mangano S, Caruso C, Spano N, Michaud L, Rizzo C. 2019. Antarctic sponges from the Terra Nova Bay (Ross Sea) host a diversified bacterial community. *Sci Rep* 9:16135.
19. Urtubia R, Gallardo P, Cárdenas CA, Lavin P, Aravena MG. 2017. First characterization of gastrointestinal culturable bacteria of Patagonian toothfish *Dissostichus eleginoides* (Nototheniidae). *Revista de biología marina y oceanografía* 52:399-404.
20. Cabrol L, Delleuze M, Szylił A, Schwob G, Quéméneur M, Misson B. 2023. Assessing the diversity of plankton-associated prokaryotes along a size-fraction gradient: A methodological evaluation. *Marine Pollution Bulletin* 197.
21. Eastman JT. 1993. Antarctic fish biology: evolution in a unique environment. Academic Press.
22. Brodeur JC, Calvo J, Clarke A, Johnston IA. 2003. Myogenic cell cycle duration in Harpagifer species with sub-Antarctic and Antarctic distributions: evidence for cold compensation. *Journal of experimental biology* 206:1011-1016.
23. Constable AJ, Melbourne-Thomas J, Corney SP, Arrigo KR, Barbraud C, Barnes DK, Bindoff NL, Boyd PW, Brandt A, Costa DP. 2014. Climate change and Southern Ocean ecosystems I: how changes in physical habitats directly affect marine biota. *Global change biology* 20:3004-3025.
24. Saravia J, Paschke K, Pontigo JP, Nualart D, Navarro JM, Vargas-Chacoff L. 2022. Effects of temperature on the innate immune response on Antarctic and sub-Antarctic fish *Harpagifer antarcticus* and *Harpagifer bispinis* challenged with two immunostimulants, LPS and Poly I: C: *In vivo* and *in vitro* approach. *Fish & Shellfish Immunology* 130:391-408.
25. Ergas M, Figueroa D, Paschke K, Urbina MA, Navarro JM, Vargas-Chacoff L. 2023. Cellulosic and microplastic fibers in the Antarctic fish *Harpagifer antarcticus* and Sub-Antarctic *Harpagifer bispinis*. *Marine Pollution Bulletin* 194:115380.
26. Vargas-Chacoff L, Martínez D, Oyarzún-Salazar R, Paschke K, Navarro J. 2021. The osmotic response capacity of the Antarctic fish *Harpagifer antarcticus* is insufficient to cope with projected temperature and salinity under climate change. *Journal of Thermal Biology* 96:102835.
27. Navarro JM, Paschke K, Ortiz A, Vargas-Chacoff L, Pardo LM, Valdivia N. 2019. The Antarctic fish *Harpagifer antarcticus* under current temperatures and salinities and future scenarios of climate change. *Progress in Oceanography* 174:37-43.

28. Carthey AJR, Blumstein DT, Gallagher RV, Tetu SG, Gillings MR, Bennett A. 2020. Conserving the holobiont. *Functional Ecology* 34:764-776.

Re: Spectrum03830-23R1 (Unveiling the co-phylogeny signal between plunderfish *Harpagifer* spp. and their gut microbiomes across the Southern Ocean)

Dear Dr. Guillaume Schwob:

Your manuscript has been accepted, and I am forwarding it to the ASM production staff for publication. Your paper will first be checked to make sure all elements meet the technical requirements. ASM staff will contact you if anything needs to be revised before copyediting and production can begin. Otherwise, you will be notified when your proofs are ready to be viewed.

Sincerely,
Konstantinos Kormas
Editor
Microbiology Spectrum

Reviewer #1 (Comments for the Author):

I kindly appreciate the great effort of the authors to solve my major concerns related with the first version of their manuscript, undoubtedly this reviewed version satisfactorily fulfilled most of them. I still have a minor comments that need to be implemented.

Minor comments

In lanes 78 and 79 the authors state that "phylosymbiosis does not imply any stable evolutionary association between a host and its microbiome along time. I clearly understand that the term phylosymbiosis could have several interpretations and the goal of the authors is to discriminate phylosymbiosis from co-phylogeny in order to justify the use of haplotypes instead of *Harpagifer* species as terminals; however, according with Lim and Bordenstein (2020) phylosymbiosis could arise also as host-microbe coevolution, codiversification and speciation, in fact a phylosymbiosis test also implies the comparison of host and associate phylogenies. It would be easy as the authors add a pair of words in lanes 79 and 80, v. gr. "However, this pattern does not imply any stable evolutionary association between a host and its microbiome along time (12) but see Lim and Bordenstein (2020)" or something like that.

The authors make an effort to clearly state the difference between phylosymbiosis and co-phylogeny in their introduction. Phylosymbiosis= congruent pattern between host phylogeny and its microbial community at an instant, co-phylogeny= parallel evolution between host and microbial symbionts suggesting a long term relationship. In fact, in lines 149 and 150 they state that their goal was to "test the hypothesis of a co-phylogenetic signal between Harpagifer species and shared members of their gut microbiomes (i.e., core microbial taxa)" As commented earlier, I could differ from their statement, but I understand that there are several research programs and each one could have a definition of what is phylosymbiosis and how must be tested, I have no problem with that and respect the authors' point of view. Nevertheless, in several sections of the discussion the authors use the term phylosymbiosis, v. gr. "a broad sampling strategy and robust statistical tests as they performed are a valid approach to detect phylosymbiosis". I read several times the discussion and I agree with the use of the term phylosymbiosis, the problem arise in the introduction where the authors put a "straitjacket" with null freedom. In fact, as I perceive the results of their work, they are dealing with a relationship at two levels, a "gross grain" phylosymbiosis relationship between the host and the GMM that could be instantaneous or a long term relationship and a "fine grain" co-phylogeny pattern between the host and *Alivibrio*, but I could be wrong with my appreciation. Whatever be the scenario, I suggest the authors clearly state what they pretend and not leave it to the discretion and interpretation of the reader. It's as simple as slightly adjusting some sentences of the introduction and goal of their work.

Please include the genebank accession numbers for new generated sequences for the species of *Harpagifer* as well as the Bioproject number for the microbiome sequences.

The name of the genera in both Figure 3 and table 3 must be in italics.

Lane 343 says "predominantly within the Gammaproteobacteria phylum...." must be "Pseudomonadota phylum (former Proteobacteria), Gammaproteobacteria is a class not a phylum.

In the table 3 says "Class Actinobacteria" and must be "Actinomycetia"

Reviewer #2 (Comments for the Author):

The authors answered with precision and clarity to my comments.